# TRAINING UNIVERSAL TEXT ENCODERS WITH PAIR RELEVANCE CLASSIFICATION LOSS

## ABSTRACT

Finetuning large language models (LLMs) using contrastive learning objectives has become the dominant approach for representation learning in general-purpose text embedding tasks. Our work seeks to enable going beyond strictly positive (or negative) pairs of text, to more fine-grained annotations that can capture the nuances of complex language tasks. We propose training text encoders with a simple pair classification loss that utilizes binary cross-entropy on relevance labels. When compared to the standard softmax-based loss for multi-class classification against multiple text alternatives, we find that training with our proposed loss improves the average score across 56 English language tasks of the Massive Text Embedding Benchmark (MTEB), while finetuning the same Meta-Llama-3-8B-Instruct model on the same mix of open datasets. Furthermore, our models excel in the Pair Classification and the Semantic Textual Similarity benchmarks, outperforming many models that are trained on more extensive data. Finally, thorough experiments using graded relevance data from TREC-DL 2023 during training demonstrate that binary cross-entropy provides generalization improvements that the softmax-based loss fails to achieve.

## 1 INTRODUCTION

Sentence embeddings form the foundation of many deep learning approaches to natural language understanding and have been extensively studied in the literature Gao et al. (2021); Reimers & Gurevych (2019). A well-designed text embedder is crucial for achieving high performance across a range of downstream tasks such as machine translation, information retrieval, and sentiment analysis Muennighoff et al. (2023). Ideally, it is desirable to have a single text encoder that maximizes the performance across all such tasks, and hence developing robust and universal sentence embedders has become a focal point in the field of natural language processing.

Designing universal sentence embedders that generalize well across diverse downstream tasks, however, remains a significant challenge. In itself, natural language is not a clear and unambiguous means of communication; it is shaped by context, speaker intention, and the specific task at hand. The contextual dependencies in language vary significantly depending on the goal (i.e. downstream task). For example, a sentence in a legal document might emphasize precision and formal terminology, whereas the same sentence in a friendly email could rely on informal tone and inferred meaning.

Generally, text encoders trained with contrastive objectives have relied on CLIP-style (Radford et al., 2021) softmax-based losses. That is, given two aligned batches of data of size $n$, a similarity matrix with dimension $n \times n$ is created by scoring all against all points across the two batches. Each row or column (or often both) is then treated as the parameters of a categorical random variable, and maximum likelihood estimation can be performed by simply placing labels at the main diagonal. While effective, this common approach is known to be highly sensitive to the choice of batch size. This is due to the dimension of the categorical random variables described above — or analogously, to the number of classes in a multi-class classification setting — being tied to the batch size. Alternative training objectives were introduced as variations of CLIP, especially so for the case of discriminative vision-language models such as SigLIP (Zhai et al., 2023). In that case, each entry in a similarity matrix obtained for aligned batches of data is treated as the parameter of a Bernoulli random variable, so maximum likelihood estimates are given by the minimizer of a binary cross-entropy (BCE) loss. Unlike the softmax setting used in CLIP, losses that operate at each entry of the similarity matrix are

less affected by the choice of the batch size, for instance, and have yielded state-of-the-art vision-text encoders. Here, we not only leverage these advantages observed in the text-vision case to improve training of text-only bi-encoders, but also take advantage of the fact that pair-wise losses offer the opportunity to use more nuanced annotations. This is especially true in cases where different types of negative or positive samples are available. For instance, if both easy and hard negative examples are available, one can reflect that in the choice of labels and, for example, put more pressure towards matching harder negatives than easier ones.

In this paper, we demonstrate the potential of using BCE loss for training universal sentence embedders. When training the same models with the same training data which use binary relevance labels (e.g. relevant, non-relevant), the BCE loss matches and even outperforms softmax-based contrastive loss in downstream tasks. When applied to tasks with graded relevance labels (e.g. relevant, partially relevant, non-relevant), BCE proves to be a much better training objective, significantly outperforming the traditional softmax-based contrastive loss. The flexibility of the BCE loss to handle more nuanced labels can more effectively tackle the ambiguity and contextual variability of natural language.

## 2 PRELIMINARIES

### 2.1 GENERALIST TEXT EMBEDDERS

The purpose of building a generalist embedder is to have a single model that can provide text embeddings for several downstream tasks including, but not limited to, retrieval, text classification, and clustering Conneau et al. (2017); Reimers & Gurevych (2019); Karpukhin et al. (2020); Muennighoff (2022). Initial approaches to train a generalist embedder relied on a multi-stage pipeline. The first stage involved training BERT-based bidirectional encoders on large-scale self-supervised data (typically comprising of 1B sentence pairs). This was then followed by fine-tuning the pretrained model on smaller-scale supervised data (typically involving 1M sentence pairs) Ni et al. (2022); Wang et al. (2022); Li et al. (2023a); Xiao et al. (2024); Su et al. (2023a); Asai et al. (2023).

More recently, many works have proposed converting decoder-only-LLMs into text-encoders using architecture modifications (BehnamGhader et al., 2024), using prompting techniques Springer et al. (2024), or by simply fine-tuning with task-specific supervised data Neelakantan et al. (2022). These approaches demonstrate superior performance than bidirectional encoders and eliminate the need for large-scale self-supervised training (Wang et al., 2023). Moreover, using LLMs for text encoding allows us to leverage their instruction-following capabilities to distinguish between different downstream tasks. Additionally, LLMs offer the flexibility to generalize to new tasks using natural language instructions For example, given the input "*Retrieve relevant documents for this user query: I need a new laptop*", the model would generate the embedding suitable for document search, whereas, for the same model, the input "*Classify the product category: I need a new laptop*" would generate embedding suitable for product classification.

In this work, we focus on LLM2Vec (BehnamGhader et al., 2024) – a three-step approach to converting any decoder-only LLM into a text encoder. Our choice is driven by LLM2Vec's open-source availability, use of only publicly available data for training, and its strong performance across a wide variety of text-based tasks on the MTEB benchmark, making it both a versatile and accessible solution for sentence embedding. LLM2Vec models are built on the hypothesis that the causal attention mechanism of LLMs is sub-optimal for text encodings. Therefore, these models first remove the causal mask of the LLM to enable better contextualization with bidirectional connections. The model is then trained with a masked next-token prediction objective to adapt it to bidirectional attention. Finally, the model can be fine-tuned with either SimCSE (Gao et al., 2021) in unsupervised settings, or with typical contrastive learning in presence of supervised data.

### 2.2 TEXT EMBEDDING BENCHMARK

The most widely-used benchmark for evaluating generalist text embedders is the Massive Text Embedding Benchmark (MTEB, Muennighoff et al. 2023), which contains several downstream embedding tasks across multiple languages. Following BehnamGhader et al. (2024), we evaluate our approach on the English MTEB benchmark, containing 56 tasks across seven task categories – retrieval, reranking, clustering, pair classification, classification, sentence similarity, and summarization.

Retrieval and reranking tasks are evaluated by encoding query and document separately using the embedder, and scoring each query-document pair using cosine similarity of their respective embeddings. This setup, typically referred to as *bi-encoder* Guu et al. (2020); Karpukhin et al. (2020); Santhanam et al. (2022), allows scaling to extremely large document collections using efficient indexing and search techniques (Johnson et al., 2019).

Clustering is evaluated by computing the v-measure score (Rosenberg & Hirschberg, 2007) on text embeddings clustered using k-means. The pair classification task involves assigning a binary label to a pair of sentences. To reformulate it as an embedding task, both sentences are first embedded, and the label is determined by applying a threshold to their similarity score (e.g., cosine similarity). Summarization and text similarity tasks follow a similar approach: for summarization, human and machine-written summaries are treated as text pairs, and sentence similarity is determined by the use of continuous labels instead of binary ones. Classification is assessed by training a logistic regression classifier on training set embeddings and testing on the validation set, with classifier performance serving as a proxy for embedding quality.

## 2.3 GRADED RELEVANCE

Softmax-based contrastive learning has been the de-facto paradigm for training dense retrievers and generalist embedders. The training loss formulation in this case relies on *binary relevance*, i.e. a boolean assignment (relevant or non-relevant) for every text pair. However, relying only on binary relevance misses out on the potentially rich supervision signal that comes from *graded relevance*, which allows for a more nuanced assessment.

Graded relevance refers to the extent to which a text pair is relevant, often quantified using an ordinal scale to capture varying levels of relevance (e.g., 0–3 or 0–5). Historically, evaluating the graded relevance assignment has been the primary method for assessing the quality of a retrieval system within the information retrieval community (Järvelin & Kekäläinen, 2000; 2002; Sakai, 2021). For instance, in the Deep Learning track of the *Text REtrieval Conference (TREC-DL) 2023* (Craswell et al., 2024), models are evaluated based on how closely their scores align with human judgment of graded relevance for query-document pairs. These graded relevance scores range from 0 to 3, representing the following levels of increasing relevancy:

- 0 = Irrelevant,
- 1 = Relevant topic, but does not contain the answer,
- 2 = Highly relevant if it contains a partial answer or if the answer is unclear, or
- 3 = Perfectly relevant, containing the exact answer.

## 3 CONTRASTIVE REPRESENTATION LEARNING FOR TEXT EMBEDDINGS

In this section, we describe our approach for training generalist text embedders using binary cross-entropy loss formulation, which is better suited to capture the nuanced assessment of graded relevance.

### 3.1 DATASET FORMAT AND SOFTMAX-BASED CONTRASTIVE LEARNING LOSS

We first introduce the standard supervised framework for training text encoders. In this setup, the training data is typically provided to the learning algorithm as tuples of related texts. Each tuple consists of an *anchor* text, a positively associated text, and possibly a hard negative text. These associations are task-specific, with each tuple assigned to a particular task objective. The resulting embeddings are often task-conditioned by combining a task description sentence with either the anchor text, in the case of asymmetric tasks, or both the anchor and associated texts. Such a combination can be as simple as text concatenation, formatted as *"{task instruction}: {text}"*.

For instance, for a symmetric task like semantic textual similarity (STS), both the anchor and associated text are generic sentences, and are combined with the instruction *"Retrieve semantically similar text"*. In contrast, for an asymmetric task like passage retrieval, where the anchor is a query and the associated text is a passage, only the query is combined with the task instruction, *"Given a query, retrieve relevant passages that answer the query"*. Details of what constitutes an anchor and

Table 1: Composition of the public portion of the E5 training dataset (Wang et al., 2023), reconstructed by Springer et al. (2024). In the task categories below, Natural Language Inference is denoted by NLI and Question-Answering as QA. Details on the instructions used for each dataset can be found at Table 5 of the Appendix.

| Dataset Name | Task Category | Meaning of anchor/associated text |
|---|---|---|
| AllNLI (Gao et al., 2021) | NLI | premise/hypothesis |
| DuReader (He et al., 2018) | Passage Retrieval | query/passage |
| ELI5 (Fan et al., 2019) | Popular Responses | forum question/user response |
| FEVER (Thorne et al., 2018) | Fact Checking | claim/document |
| HotpotQA (Yang et al., 2018) | Passage Retrieval for Multi-Hop QA | query/passage |
| Miracl (Zhang et al., 2023) | Passage Retrieval | query/passage |
| MrTydi (Zhang et al., 2021) | Passage Retrieval | query/passage |
| MSMARCO (Bajaj et al., 2018) | Passage Retrieval | query/document or passage |
| Natural Questions (Kwiatkowski et al., 2019) | Passage Retrieval | query/Wikipedia article |
| Quora Duplicates (DataCanary et al., 2017) | Duplicates Classification | forum question/forum question |
| SQuAD (Rajpurkar et al., 2016) | Passage Retrieval | query/Wikipedia article |
| T2Ranking (Xie et al., 2023) | Passage Retrieval | query/web passage |
| TriviaQA (Joshi et al., 2017) | Passage Retrieval | query/Wikipedia article |

an associated text for each task in our training dataset can be found at Table 1. The full set of task instructions for each dataset is provided in Table 5 of the Appendix.

Typically, text encoders are trained with softmax-based contrastive learning loss. Let $f$ be the text encoder we would like to train, which, in general, is a neural network that takes text as input and outputs a fixed-dimensional vector. During training, the encoder is presented with batches of text tuples $\mathcal{B} = \{(x_k, y_k^+, y_k^-)\}_{k=1}^B$. Let $\mathcal{X} = \{x_k\}_{k=1}^B$ be the batch of anchor texts, and $\mathcal{Y}^+ = \{y_k^+\}_{k=1}^B$ and $\mathcal{Y}^- = \{y_k^-\}_{k=1}^B$ be the associated positive and (hard) negative text batches respectively. To train with the softmax-based contrastive learning task, we first compute the normalized embeddings for the anchor, $\boldsymbol{x} = \frac{f(x)}{\|f(x)\|}$, the positive examples, $\boldsymbol{y}^+ = \frac{f(y^+)}{\|f(y^+)\|}$, and the negative examples, $\boldsymbol{y}^- = \frac{f(y^-)}{\|f(y^-)\|}$. The encoder is then trained by minimizing the following loss function, where $s(x,y) = \alpha\, \boldsymbol{x}^\top \boldsymbol{y}$:

$$\frac{1}{B} \sum_{x_k \in \mathcal{X}} - \log \frac{\exp(s(x_k, y_k^+))}{\sum_{y \in \mathcal{Y}^+ \cup \mathcal{Y}^-} \exp(s(x_k, y))} \tag{1}$$

This loss function corresponds to the negative log-likelihood of classifying each anchor text $x_k$ into one of $2B$ possible classes, where each class represents an associated text from the batch. Given the normalized embedding $\boldsymbol{x}_k$ for the anchor text, the $2B$ logits are computed by taking the dot product between $\boldsymbol{x}_k$ and the normalized embeddings $\boldsymbol{y}$ of all associated texts in the batch[1]. These dot products are then multiplied by a scalar hyperparameter $\alpha > 0$, which acts as an inverse temperature.

The target class for the anchor text $x_k$ is always determined by its positively associated text $y_k^+$. In contrast, its own (hard) negative example, along with the associated texts and (hard) negative examples from other anchors in the batch, define the alternate (negative) classes. Training with *in-batch* negatives is essential for achieving strong performance on downstream tasks (Karpukhin et al., 2020), particularly in the absence of hard negative texts. This approach provides the model with additional data to effectively distinguish between relevant and irrelevant texts. In the self-supervised learning literature, this loss is commonly referred to as the InfoNCE objective (Oord et al., 2018).

**The difficulty of measuring relevance with Softmax-based contrastive learning.** Using graded relevance scores for text pairs using softmax-based contrastive learning presents significant challenges. In this work, we explore how model training can be improved if we have access to a training set with graded relevance scores, instead of just binary ones, for each pair of anchor $x$ and associated text $y$. We argue that this task is not straightforward under softmax-based contrastive learning because the multi-class classification problem defined in Equation (1) targets a one-hot probability vector, where the 1 is located at the index of the positive associated text.

To illustrate this challenge, consider a scenario where in-batch negatives consistently have a relevance score of 0 (since they are random texts from the dataset). Naively renormalizing the relevance scores

---

[1]The dot product of $L^2$-normalized vectors is also known as cosine similarity.

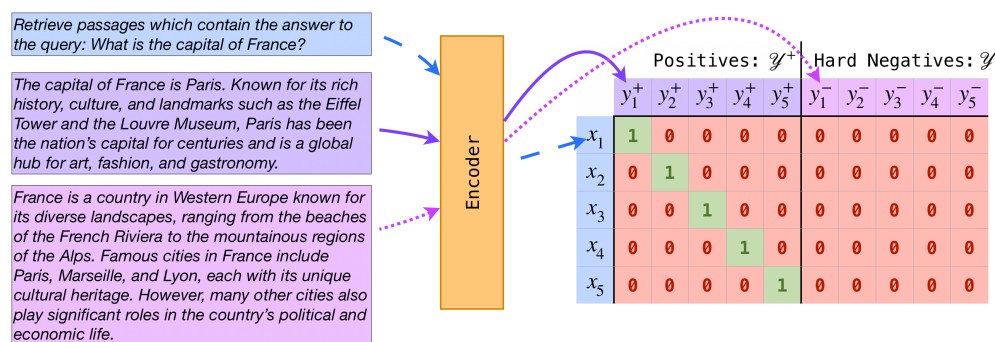

Figure 1: Training text encoders with the binary cross-entropy (BCE) loss on pair relevance scores. The dataset provides with anchor texts to be contrasted against a positively associated text and potentially a hard negative one. Using in-batch negatives makes efficient use of the encoded texts in a batch by assuming associated texts of other anchors as negatives. In this example, the label for positively associated pairs of texts in the BCE loss is 1, while for negatives is 0.

of the $2B$ possible classes into a target probability vector fails to provide a more fine-grained training score, because this would result in a one-hot target vector regardless of whether the relevance score for the positive pair is 1 or 3. Therefore, we propose a solution that leverages graded relevance scores with an alternative contrastive loss function. This approach focuses on directly classifying the relevance of a pair of texts by minimizing the binary cross-entropy loss on the relevance labels, rather than employing the cross-entropy loss on the possible associated texts.

## 3.2 PREDICTING PAIR RELEVANCE SCORES AS A REPRESENTATION LEARNING TASK

Training an encoder $f$ using binary cross-entropy loss requires access to labels $z \in [0,1]$ for each pair of anchor and associated text $(x, y)$. In its simplest form, this label is binary, indicating whether the pair is positive ($z = 1$) or negative ($z = 0$). However, it can also take on intermediate values, representing degrees of relevance. For generality, we define $z$ as a function of a text pair, $z : (x, y) \mapsto [0, 1]$. Let $\sigma$ denote the sigmoid (logistic) function, $\sigma(t) = \frac{1}{1+\exp(-t)}$. As before, batches of text tuples $\mathcal{B}$ are presented to the encoder during training. Assuming that the labeling function returns zero for in-batch negatives, that is $z(x_k, y) = 0$ for all $y \in \mathcal{Y}^+ \cup \mathcal{Y}^- \setminus \{y_k^+\}$, the encoder $f$ is trained to minimize the following function, where $s(x, y) = \alpha \, \boldsymbol{x}^\top \boldsymbol{y} + \beta$:

$$-\frac{1}{B} \sum_{x_k \in \mathcal{X}} \sum_{y \in \mathcal{Y}^+ \cup \mathcal{Y}^-} z(x_k, y) \log \sigma\big(s(x_k, y)\big) + \big(1 - z(x_k, y)\big) \log \sigma\big(-s(x_k, y)\big) \quad (2)$$

Compared to Equation (1), the formula for the logits of the binary cross-entropy loss introduces an additional logit bias term $\beta$, while the logit scale is controlled by a parameter $\alpha > 0$. The behavior of this loss function depends on the label $z(x_k, y)$. When $z(x_k, y) > 0.5$, we want the sigmoid to output a value closer to 1, implying that the normalized embeddings $\boldsymbol{x}_k$ and $\boldsymbol{y}$ need to be more aligned. Conversely, when $z(x_k, y) \leq 0.5$, the optimization process drives the sigmoid output towards zero, encouraging the dot product between the embeddings to be negative. The overall behavior of this loss over a batch $\mathcal{B}$ and the labels assigned in the case of binary $z$ is summarized in Figure 1.

**The role of logit bias $\beta$.** When using in-batch negatives, the underlying binary classification problem becomes label-imbalanced. This imbalance arises because the model encounters one positive instance for every $2B - 1$ negative instances at each training step. This can pose challenges, particularly with large training batch sizes, as it becomes progressively easier for the encoder $f$ to minimize the loss by simply predicting the negative class for all pairs. Consistent with the observations made in SigLIP, we also observe that the downstream task performance is influenced by the initialization value of the additive logit bias $\beta$, which we analyze further in Section 5.

We understand the logit bias $\beta$ in Equation (2) as a method for addressing the label-imbalance problem. By carefully selecting $\beta$, we adjust the resulting logits to account for the spurious marginal

statistics on the random label $z$, regardless of the input pair $(x, y)$. This approach allows us to initialize our optimization problem from a point where the only way for making progress is by encoding information useful for predicting $z$ in the dot product $\boldsymbol{x}^T \boldsymbol{y}$, independent of the training distribution of $z$. This idea aligns with the logit adjustment literature on long-tailed multi-class classification (Menon et al., 2021), where a similar logit bias intervention is removed during inference to unveil a more robust classifier. In particular, Menon et al. (2021) modify the unnormalized logits of a K-class classifier by adding a fixed estimate for the log-likelihood of each one of the classes to the corresponding class logit predicted by the network. By training this logit-adjusted classifier with the regular cross-entropy loss and removing the logit bias at the end of training, they show that the remaining trained classifier is robust to shifts in the label distribution.

Similarly, we would like our binary classifier to accurately assess the relevance of text pairs, regardless of the training distribution of $z$, which is defined as $p_{\text{data}}(\mathbf{z} = 1) = \frac{1}{2B}$ when we use in-batch negatives. We thus propose the following formulation for the logits of the binary cross-entropy loss:

$$\mathrm{s}^{(\text{adjusted})}(x, y) = \alpha \, \boldsymbol{x}^\top \boldsymbol{y} + \tau \log \frac{p_{\text{data}}(\mathbf{z} = 1)}{p_{\text{data}}(\mathbf{z} = 0)} \,, \tag{3}$$

where $\tau \approx 1$ corresponds to a scalar tuned to account for model calibration errors (Guo et al., 2017). Notice that under our proposal, $\beta = \tau \log \frac{p_{\text{data}}(\mathbf{z}=1)}{p_{\text{data}}(\mathbf{z}=0)}$. As a numerical example, we see that the best results for SigLIP are recorded for batch size $32, 768$ (Zhai et al., 2023, see Figure 2). In this case, $p_{\text{data}}(\mathbf{z} = 1) = \frac{1}{B} = \frac{1}{32768}$, as no hard negative examples are used. Substituting this value into our formula for $\beta$, we obtain $\beta_{\text{SigLIP}} = \log \frac{\frac{1}{B}}{1 - \frac{1}{B}} = -\log(B - 1) = -10.397$, which closely matches the reported best hyperparameter value of $-10$ for $\beta$.

### 3.3 Enabling Training with Graded Pair Relevance Scores

Suppose we have access to a training set containing pairs of anchor and associated texts $(x, y)$, along with their respective relevance scores $r$. As introduced in Section 2.3, $r$ values typically take ordinal values ranging from 0 to some maximum preference value $R$. If we could effectively translate these scores into probability targets in $[0, 1]$, we could leverage the binary cross-entropy loss for training. We propose using the following:

$$z(x, y) = c + (1 - c)\frac{r(x, y)}{R} \,, \text{if } r(x, y) > 0 \text{ otherwise } 0, \tag{4}$$

as a simple protocol for achieving this, where $c \in (0, 1)$ serves as a cutoff value. To illustrate, consider a numerical example for TREC-DL 2023 assuming $c = 0.7$. This transforms relevance $r$ of 1 to $z$ value of 0.8, 2 to 0.9 and 3 to 1.0. Then, given a batch $\mathcal{B}_{\text{graded}} = \{(x_k, y_k, z_k)\}_{k=1}^{B}$ with graded labels we can use the following binary cross-entropy loss to learn this more fine-grained information:

$$-\frac{1}{B} \sum_{x_k, y_k, z_k \in \mathcal{B}_{\text{graded}}} z_k \log \sigma\big(s(x_k, y_k)\big) + (1 - z_k) \log \sigma\big(-s(x_k, y_k)\big) \,, \tag{5}$$

where we use the logit function from Equation (3) and binarized values for $z$ in order to estimate $p_{\text{data}}(\mathbf{z} = 1)$. Notice that under the formulation of Equation (5) we do not make use of in-batch negatives. Instead, negatives are provided by the dataset whenever $z_k = 0$.

## 4 Training Setup

We base our experiments mostly on LLM2Vec (BehnamGhader et al., 2024), which is a strong open-source sentence embedder trained with publicly-available and permissive datasets. For fair comparison, we use the same training data as LLM2Vec, which includes the public portion of the E5 dataset (Wang et al., 2022) reconstructed by Springer et al. (2024). In Table 1 we describe the composition of training sets and their corresponding tasks. In Table 5 of the Appendix, we provide the list of instructions we used to augment the anchor texts, and also their associated text in the case of symmetric tasks. We use the same task instructions as previous works Wang et al. (2023); Springer et al. (2024); BehnamGhader et al. (2024). In addition to comprising public permissible datasets, this training data mix has minimal overlap with MTEB tasks, enabling us to evaluate the robustness of our models on domains beyond the training set.

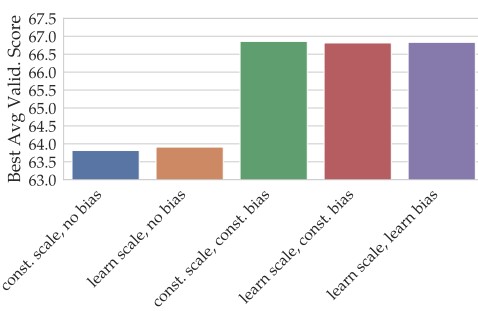 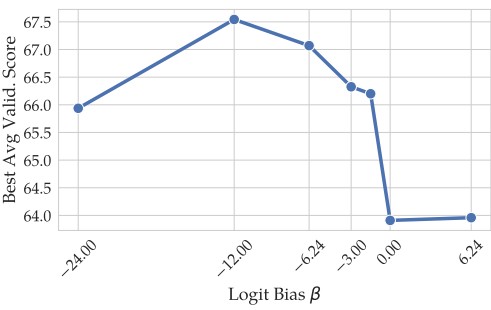

(a) Ablation on learnability of logit scale and bias.      (b) Effect of logit bias value on performance.

Figure 2: Use of logit bias and determining its value correctly is essential for having good performance with the binary cross-entropy loss. We perform experiments with batch size $B = 256$ and confirm that Equation (3) provides with a good default value for the logit bias $\beta = -\log(2B - 1) \approx -6.24$.

The LLM2Vec approach for converting LLMs into encoders consists of three steps: (1) enabling bidirectional connections, (2) using masked next token prediction (MNTP) to adapt these connections, and (3) applying supervised contrastive learning. In our setup, we initialize models with the MNTP checkpoint of LLM2Vec models to fairly compare the performance of softmax-based contrastive loss against our proposed binary cross-entropy loss for supervised training. Specifically, we use Meta-LLaMA-3-8B-Instruct (hereafter referred to as **Meta-LLaMA-3**), and Sheared-LLaMA-1.3B (hereafter referred to as **S-LLaMA**) to evaluate performance across different model sizes.

We also provide experiments with Mistral-7B-Instruct (hereafter referred to as **Mistral**) and Qwen2-7B-Instruct (hereafter referred to as **Qwen2**), for which we adopt a simpler conversion strategy than LLM2Vec. In particular, we initialize from pretrained foundational models and we simply enable bidirectional connections between tokens during finetuning with contrastive learning.

**Training details.** We employ the Adam optimizer (Kingma & Ba, 2015) with $\beta_1 = 0.9$, $\beta_2 = 0.995$. We apply no weight decay. For the learning rate scheduler, we adopt a linear warmup over 300 steps, followed by a linear decay until 10,000 steps. The logit scale is tuned. For the binary cross-entropy loss, we find the optimal value to be 20, which corresponds to a temperature of 0.05. In our generalist text encoder experiments, we make use of in-batch negatives and apply the loss function described in Equation (2). We implement task-conditioned sampling for batching, in which all the samples in a batch belong to the same task. Doing so improves the quality of in-batch negatives as all the associated text belong to the same domain. We describe training in greater detail in Section 7 of the Appendix, along with details about model selection where we define the validation score we use in our studies, as well as for hyperparameter search.

## 5 EXPERIMENTS ON MASSIVE TEXT EMBEDDING BENCHMARK

**Effect of batch size.** Training batch size is typically a sensitive hyperparameter when finetuning for a generalist text encoder, as it implies the number of negatives used per anchor text at each training step, in the case of in-batch negatives. As we observe in Figure 3a, we find that our proposed loss performs well with smaller batch sizes. This is especially true for the Meta-LLaMA-3 model for which the best model is found while training with batch size 256. As we analyze in Section 7, we attribute this feature of BCE losses to the particular form of the gradients for the negative pairs which, in contrast to the softmax-based loss, does not induce a competition among the negative pairs for contributing to the total gradient.

**Ablation of logit parameters.** We conduct our ablation of logit scale $\alpha$ and the logit bias $\beta$ of Equation (2) on S-LLaMA models with batch size 256. Both CLIP (Radford et al., 2021) and SigLIP (Zhai et al., 2023) utilize a learnable $\alpha$, and at the same time SigLIP and our method introduce an extra logit bias $\beta$. In Figure 2a, we perform an ablation experiment among the following 5 cases: (1) *constant $\alpha$ and no $\beta$* which corresponds to the typical logit formulation used in softmax-based contrastive losses, where only a scaled dot product of normalized embeddings is used. (2) *learnable $\alpha$*

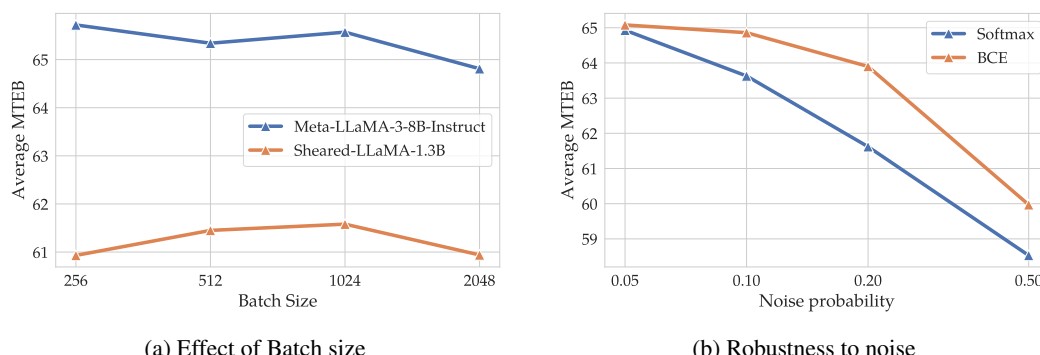

(a) Effect of Batch size        (b) Robustness to noise

Figure 3: **Left:** Various models trained with different batch sizes using our binary cross-entropy loss. **Right:** Model performance after preprocessing training dataset by flipping positive pairs with hard negatives according to a noise probability. On both graphs, we display the average performance on the full 56-task English MTEB.

*and no $\beta$ corresponds* to the setting of CLIP and we observe a slight improvement over case (1). The benefits become significant, however, once we introduce $\beta$. Case (3) demonstrates the performance of using *constants $\alpha$ and $\beta$*, which seems to achieve the maximum performance with respect to the validation score over case (4) - where *only $\alpha$ is learnable* - and case (5) - where *both $\alpha$ and $\beta$ are learnt* like SigLIP.

**Effect of logit bias.** Using a constant logit scale and bias, we investigate the effect of the logit bias, treated as a hyperparameter, by training S-LLaMA models with a batch size of 256. As shown in Figure 2b, we observe that the performance on our validation score peaks at a logit bias value of $-12$. The second-best score is achieved at $-6.24$, which is the value predicted by our proposal in Equation (3), based on a batch size of 256 with hard negatives. The discrepancy between the predicted and optimal hyperparameter values may be due to model calibration errors (Guo et al., 2017). Another factor is that our model is not initialized to exactly predict according to the marginal training distribution of labels, especially since we finetune a pre-trained model to create the general text encoder. To conclude, while a small hyperparameter search around the predicted value is recommended, the predicted value for the logit bias can serve as a competitive default.

**Effect of weakly-supervised dataset noise.** We perform an ablation study with respect to levels of artificial noise introduced to the open-source E5 training dataset. In particular, we specify a probability of flipping the label between a designated positive pair $(x, y^+)$ and a hard negative pair $(x, y^-)$, and we sample the flip variable independently for every triplet $(x, y^+, y^-)$ in the dataset prior to training. Then, we train Meta-LLaMA-3 models with the BCE and softmax-based losses and we report the average 56-task MTEB score of the best models found during training. In Figure 3b we observe that BCE outperforms the softmax variant for all noise levels $p_{\text{noise}} \in \{0.05, 0.1, 0.2, 0.5\}$. We hypothesize that BCE achieves better robustness to weakly-supervised dataset noise, because the effect of noise is more diluted in the gradients of the BCE objective. Suppose the binary label case with hard negatives. At each training step the model encounters a batch of size $B$, the BCE-loss then is the average of losses from $2B^2$ binary classification predictions, while softmax-based loss is the average of losses from $B$ multi-class classification predictions. Assuming that the in-batch negatives are indeed true negatives: If one pair of texts in the batch is mislabeled, then for the softmax-loss this means that the $\frac{1}{B}$ predictions is optimized towards a false target, whereas for the BCE-loss $\frac{1}{B^2}$ predictions is optimized towards a false target.

**Full MTEB results.** We train the LLM2Vec models on the public E5 dataset with binary cross entropy loss using the default values of the logit bias. After selecting the best models across a training session based on their validation score, we evaluate them across the complete suite of 56 tasks from the English MTEB. We contrast our approach with softmax-based contrastive finetuning and with other methods that utilize publicly available and permissible data in Table 2. We also provide a comparison with current top entries in the MTEB leaderboard (as of October 1, 2024) in Table 7.

First, we compare our approach to the softmax-based contrastive finetuning alternatives of the models we consider. We find that our training method leads to improvements in 4 out of 7 task categories

Table 2: Results on the 56 tasks of the English MTEB benchmark. Retrieval (Retr.) performance is measure using NDCG@10. Reranking (Rerank.) performance uses Mean Average Precision. Clustering (Clust.) uses the V-measure. Pair classification (Pair) uses Average Precision (AP) based on the models similarity metric. Classification (Class.) uses accuracy. Semantic Textual Similarity (STS) tasks and Summarization (Summ.) use the Spearman correlation based on the model's similarity metric. Finally, the average (Avg) is computed by averaging the scores of all individual tasks together. $^\star$versions of models that were finetuned with public MEDI2BGE and E5 datasets.

| Categories → # of datasets → | Retr. 15 | Rerank. 4 | Clust. 11 | Pair 3 | Class. 12 | STS 10 | Summ. 1 | Avg 56 |
|---|---|---|---|---|---|---|---|---|
| INSTRUCTOR-XL [45] | 49.26 | 57.29 | 44.74 | 86.62 | 73.12 | 83.06 | **32.32** | 61.79 |
| BGE-LARGE-EN-V1.5 [49] | 54.29 | 60.03 | 46.08 | 87.12 | 75.97 | 83.11 | 31.61 | 64.23 |
| GRITLM-7B$^\star$ [31] | 53.10 | **61.30** | **48.90** | 86.90 | 77.00 | 82.80 | 29.40 | 64.70 |
| E5-MISTRAL-7B-INSTRUCT$^\star$ [48] | 52.78 | 60.38 | 47.78 | 88.47 | 76.80 | 83.77 | 31.90 | 64.56 |
| ECHO-MISTRAL-7B-INSTRUCT [43] | 55.52 | 58.14 | 46.32 | 87.34 | **77.43** | 82.56 | 30.73 | 64.68 |
| LLM2VEC SHEARED-LLAMA-1.3B | | | | | | | | |
| with softmax-based loss [3] | 51.44 | 55.38 | 43.57 | 86.20 | 72.21 | 83.58 | 30.01 | 61.85 |
| with BCE loss (ours) | 52.28 | 54.16 | 43.59 | 86.61 | 70.56 | 83.10 | 30.79 | 61.58 |
| LLM2VEC META-LLAMA-3-8B-INSTRUCT | | | | | | | | |
| with softmax-based loss [3] | 56.63 | 59.68 | 46.45 | 87.80 | 75.92 | 83.58 | 30.94 | 65.01 |
| with BCE loss (ours) | 57.38 | 59.29 | 47.18 | **88.85** | 76.05 | **85.31** | 31.19 | 65.72 |
| MISTRAL-7B-INSTRUCT | | | | | | | | |
| LLM2VEC with softmax-based loss [3] | 55.99 | 58.42 | 45.54 | 87.99 | 76.63 | 84.09 | 29.96 | 64.80 |
| with BCE loss (ours) | 57.43 | 58.64 | 46.34 | 88.53 | 74.31 | 84.90 | 30.14 | 65.04 |
| QWEN2-7B-INSTRUCT | | | | | | | | |
| with softmax-based loss | 57.67 | 59.62 | 49.37 | 88.67 | 75.85 | 84.95 | 31.30 | 66.13 |
| with BCE loss (ours) | **57.84** | 59.59 | 48.67 | 88.76 | 76.84 | 84.67 | 31.73 | **66.22** |

for S-LLaMA 6 out of 7 categories for Meta-LLaMA-3, 6 out of 7 categories for Mistral, and 4 out of 7 for Qwen2. In terms of average performance across the 56 tasks, only our S-LLaMA model falls slightly short of the softmax-based baseline, with a relative performance difference of -0.44%. In contrast, our Meta-LLaMA-3, Mistral and Qwen2 models surpass their corresponding baselines, achieving a 1.10%, 0.37% and 0.14% improvement in relative performance respectively. Our results hint that our training method scales better than the softmax-based baseline with the increase in number of model parameters. We report the individual task scores for S-LLaMA and Meta-LLaMA-3 trained with binary cross-entropy in Table 9.

Overall, we find that training generalist text encoders with binary cross-entropy losses instead of softmax-based ones leads to models that perform on-par or better across a broad spectrum of downstream tasks, while being more resistant to weakly-supervised dataset noise and more robust to the choice of batch size used during training.

## 6 EXPERIMENTS ON GRADED RELEVANCE SCORES

In this section, we assess whether our proposed binary cross entropy loss function effectively improves training for graded relevance tasks. To this end, we experiment with further finetuning of general text encoders on TREC-DL Document Retrieval Challenge data (Craswell et al., 2024). We evaluate the models on TREC-DL 2023, which contains graded relevance judgements of query-document pairs from MSMARCOv2. In particular, we gather the validation and test set from TREC-DL 2023 and construct the training set from TREC-DL datasets from years preceding 2023. We make sure that there is no overlap in either the queries or the documents between train and validation/test splits. Further details about the dataset creation and its statistics, can be found in the Appendix.

All splits contain graded relevance scores for pairs, ranging from 0 to 3, as described in Section 2.3. We convert the training dataset into targets $z$ via the heuristic in Equation (4) and then train LLM2Vec Meta-LLaMA-3 models using the objective in Equation (5). For all training trials, we initialize new LORA adapters and use batch size 256. For all other training details, we follow the same training configuration as in Section 4. We select the best model for each training trial according to the maximum NDCG@100 on the validation set, and report results on the test set.

Beyond finetuning with BCE on graded relevancies, we also consider the following baselines:

Table 3: Results on the test set of the TREC-DL dataset; mean $\pm$ std is computed over 5 seeds.

|  | NDCG@10 | NDCG@20 | NDCG@50 | NDCG@100 |
|---|---|---|---|---|
| Pretrained with softmax loss [Equation (1)] | 53.68 | 55.01 | 58.32 | 65.98 |
| + Softmax ft on binarized relevancies | 55.70 | 56.76 | 61.27 | 68.28 |
| + IW Softmax ft on binarized relevancies | 50.34 | 54.37 | 59.61 | 66.50 |
| + BCE ft on binarized relevancies | 30.70 | 32.12 | 38.98 | 49.53 |
| + BCE ft on graded relevancies | 31.34 | 33.14 | 39.29 | 50.10 |
| Pretrained with BCE loss [Equation (2)] | 55.60 | 56.55 | 61.45 | 68.47 |
| + Softmax ft on binarized relevancies | 55.78 | 56.67 | 61.71 | 69.18 |
| + IW Softmax ft on binarized relevancies | 56.79 | 57.00 | 61.92 | 69.14 |
| + BCE ft on binarized relevancies | 57.56 | 57.77 | 63.03 | 69.48 |
| + BCE ft on graded relevancies | **58.55**$_{\pm 0.61}$ | **58.49**$_{\pm 0.44}$ | **63.84**$_{\pm 0.28}$ | **70.34**$_{\pm 0.17}$ |

1. Softmax-based contrastive loss on binarized relevancies: This is a natural baseline, in which at every training step we train on 128 positively associated query-document pairs, and 128 negative pairs with the same queries. We train using in-batch negatives.

2. Importance-weighted (IW) softmax-based contrastive loss on binarized relevancies: We craft this baseline as a way to highlight the difficulty of adapting softmax-based losses for graded relevance scores. The only difference with case (1) is that we use importance weights for the negative log-likelihoods, instead of simply averaging them across a batch. We compute the importance weights by normalizing the targets for the graded relevancies, as computed by Equation (4).

3. Binary cross-entropy loss on binarized relevancies: In this case, we reduce the graded scores in the training set binary scores of 0 and 1, with 1 assigned to all positive scores. This way, we aim to emphasize the benefit of using more fine-grained training signals whenever they are available.

In Table 3, we report NDCG scores of finetuned models at 10, 20, 50, and 100 top-ranked documents in the test set. For the BCE-pretrained model, we find that finetuning with graded relevance scores using our proposed loss results in the best performing model out of all the options (70.34 NDCG@100). In particular, finetuning with BCE on graded targets outperforms both the BCE baseline that uses binarized targets (69.48 NDCG@100) and the softmax-based alternatives (69.14 NDCG@100 with IW and 69.18 without).

While all the finetuning options improve upon the base model for BCE-pretrained model, the same does not hold for softmax-pretrained model. In this case, we find that the standard softmax-based contrastive recipe is the only finetuning strategy that improves upon the pretrained model (68.28 vs 65.98 NDCG@100). Trying to utilize importance-weighting to inform the model about graded relevance scores shows varied performance across different NDCG cutoffs. Finally, using BCE objectives to finetune the softmax-pretrained model significantly degrades the performance. Overall, models trained with our proposed BCE loss demonstrates stronger generalization and adaptation to graded relevance datasets compared to those trained with softmax contrastive loss.

## 7 CONCLUSION

In this work we explore training universal text encoders with binary cross-entropy loss. We show that the resulting models match or outperform those trained with the typical softmax-based contrastive loss (Section 5) on general text embedding downstream tasks. Furthermore, our proposal enables the effective use of graded relevance scores, commonly used by the information retrieval community, to provide more fine-grained training signal. This leads to significant improvements in nuanced evaluations that go beyond a binary determination of query-document relevance (Section 6). In contrast, training with softmax-based contrastive loss does not lead to similar advantages.

As models continue to advance on existing benchmarks, there is an increasing need for more nuanced performance measures to capture differences in their capabilities. One approach is to introduce graded relevance scores for text embedding tasks that can accommodate these distinctions. Our proposal will prove useful to future generations of text encoders aiming to provide with a more detailed assessment of textual relevance.

REPRODUCIBILITY STATEMENT

In the spirit of ensuring reproducibility, we have provided clear and detailed instructions for replicating the results presented in this paper. Our model selection process, hyperparameter search and the final values are detailed in Section 4 and in the Appendix. The E5 dataset used in this paper is publicly available. In addition, we provide the splits we have used for the TREC-DL dataset. Code is made available at `https://anonymous.4open.science/r/sigcse-0E2C/README.md`.

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

APPENDIX

EXTENDED RELATED WORK

**Past work on representation learning with binary cross-entropy losses.** To our knowledge, binary cross-entropy loss in self-supervised representation learning was first explored by Hjelm et al. (2019). Their work introduces Deep InfoMax (DIM), a framework for representation learning that maximizes mutual information between different views of the same image. In its Jensen-Shannon mutual information formulation (Hjelm et al., 2019, See Equation 4, Section 3.1), the DIM loss functions as a binary cross-entropy loss and the logits are computed from pairs of views: pairs derived from the same image are labeled as positive, while pairs from different images are labeled as negative.

More recently, Zhai et al. (2023) introduced SigLIP, a framework that pretrains vision-language encoders by aligning image embeddings with their corresponding caption embeddings. Compared to its softmax-based anchor classification alternative (Radford et al., 2021, CLIP), pretraining with binary cross-entropy loss in SigLIP has demonstrated better scaling of downstream task performance as the number of in-batch negatives increases. In our work, we adapt the SigLIP objective to fine-tune large language models (LLMs) (BehnamGhader et al., 2024) for general-purpose text embedding tasks.

**Pointwise regression-based ranking.** While our focus is on general-purpose text embeddings, related work can be found in training cross-encoding pointwise ranking models for information retrieval. Zou et al. (2021) mobilize training objectives on graded relevance labels using regression-based losses, however they do not prescribe a way to deal with the imbalances of the training relevance data. Our work fills that gap by allowing pointwise training on imbalanced relevance data.

TRAINING AND EVALUATION DETAILS

Table 4: Selection of validation datasets from various tasks.

| Task Category | Datasets in Validation Score |
|---|---|
| Clustering | BiorxivClusteringS2S, TwentyNewsgroupsClustering |
| Classification | Banking77Classification, EmotionClassification |
| Pair Classification | SprintDuplicateQuestions |
| Retrieval | DBPedia, HotpotQA, FiQA2018, FEVER, QuoraRetrieval |
| Reranking | StackOverflowDupQuestions |
| Sentence Textual Similarity | STSBenchmark, SICK-R |

Table 5: Instructions used for each of the E5 datasets.

| Dataset | Instruction(s) |
|---|---|
| NLI | Given a premise, retrieve a hypothesis that is entailed by the premise |
| | Retrieve semantically similar text |
| DuReader | Given a Chinese search query, retrieve web passages that answer the question |
| ELI5 | Provided a user question, retrieve the highest voted answers on Reddit ELI5 forum |
| FEVER | Given a claim, retrieve documents that support or refute the claim |
| HotpotQA | Given a multi-hop question, retrieve documents that can help answer the question |
| MIRACL | Given a question, retrieve Wikipedia passages that answer the question |
| MrTyDi | Given a question, retrieve Wikipedia passages that answer the question |
| MSMARCO Passage | Given a web search query, retrieve relevant passages that answer the query |
| MSMARCO Document | Given a web search query, retrieve relevant documents that answer the query |
| NQ | Given a question, retrieve Wikipedia passages that answer the question |
| QuoraDuplicates | Given a question, retrieve questions that are semantically equivalent to the given question |
| | Find questions that have the same meaning as the input question |
| SQuAD | Retrieve Wikipedia passages that answer the question |
| T2Ranking | Given a Chinese search query, retrieve web passages that answer the question |
| TriviaQA | Retrieve Wikipedia passages that answer the question |

**Further training details.** The learning rate is initially set to a base batch size of 64 and then scaled according to the square root of the ratio between the total batch size across devices and the base

batch size (Malladi et al., 2022), expressed as $\sqrt{\frac{total\_batch\_size}{64}}$. We apply gradient checkpointing and Parameter-Efficient Fine-Tuning (PEFT), specifically using LORA (Hu et al., 2022) on the attention module weights with $r = 16$ and $\alpha = 32$, along with a dropout rate of 0.05. For our experiments, we search for batch sizes among $\{256, 512, 1024, 2048\}$ for S-LLaMA and we fit a batch size of 128 per 80GB NVIDIA H100 GPU, scaling the number of GPUs according to the experimental needs. For the larger models (Meta-LLaMA-3, Mistral and Qwen2) we search among $\{256, 512, 1024, 2048, 4096\}$ and we fit a batch size of 128 per GPU.

The reported generalist BCE models are trained with batch sizes 256 for Meta-LLaMA-3, 512 for Qwen2, and 1024 for Mistral. For the Qwen2model we are also warming up the logit bias from -25 to the proposed value along with the learning rate. We also train a Qwen2 with a softmax-based loss with batch size 1024.

**Model Selection.** For model selection, we derive a validation score by using 0-1 tasks from each task category on MTEB, except for retrieval tasks where we subsample 256 queries and up to 131,072 passages from the development sets of DBPedia, HotpotQA, FiQA2018, FEVER, and QuoraRetrieval. We do this because evaluating many retrieval tasks is time-consuming due to the large corpus size, often in the millions of documents. To mitigate the variance in average validation scores from the subsampled retrieval dev sets, we also increase the number of tasks measured during validation to 5. Finally, we stratify the average scores from each task category in the validation set according to weights that reflect the distribution of tasks in the final benchmark and aggregate them into a single score (see Table 4). This average validation score is used both for selecting the best model during a single training run and for hyperparameter search and analysis, as presented in the experimental results section.

**Validation Score.** First, we compute the average score per task category and then we weight the means so that the aggregate simulates the distributions of tasks across the full MTEB. We hope that this way we receive a score that is fast to compute and at the same time remains somewhat indicative of the downstream performance on the full benchmark, without 'overfitting' to it. We do this by only using 0-2 datasets from each task category and utilizing available validation and development splits. The selection of dataset in the mix of the validation score can be found at Table 4.

**E5 and MTEB instructions.** As we have mentioned at Section 4, we use the public portion of the E5 dataset for training our generalist text encoders. We introduce it at Section 3.1 and we provide with an overview of it at Table 1. Training however generalist encoders can leverage an extra conditioning signal to improve generalization across tasks (Su et al., 2023a). This conditioning signals comes in the form of a task-specific instruction, which can be combined with the anchor text, and potentially with its associated texts. At Table 5, we enlist the instructions that we use during training with each of the different dataset contained in the mix. At Table 6, we enlist the instructions that we use during evaluation with each of the different tasks contained in MTEB.

Table 6: Instructions used for evaluation on the MTEB benchmark. "STS*" refers to all the STS tasks.

| Task Name | Instruction |
|---|---|
| AmazonCounterfactualClassif. | Classify a given Amazon customer review text as either counterfactual or not-counterfactual |
| AmazonPolarityClassification | Classify Amazon reviews into positive or negative sentiment |
| AmazonReviewsClassification | Classify the given Amazon review into its appropriate rating category |
| Banking77Classification | Given a online banking query, find the corresponding intents |
| EmotionClassification | Classify the emotion expressed in the given Twitter message into one of the six emotions: anger, fear, joy, love, sadness, and surprise |
| ImdbClassification | Classify the sentiment expressed in the given movie review text from the IMDB dataset |
| MassiveIntentClassification | Given a user utterance as query, find the user intents |
| MassiveScenarioClassification | Given a user utterance as query, find the user scenarios |
| MTOPDomainClassification | Classify the intent domain of the given utterance in task-oriented conversation |
| MTOPIntentClassification | Classify the intent of the given utterance in task-oriented conversation |
| ToxicConversationsClassif. | Classify the given comments as either toxic or not toxic |
| TweetSentimentClassification | Classify the sentiment of a given tweet as either positive, negative, or neutral |
| ArxivClusteringP2P | Identify the main and secondary category of Arxiv papers based on the titles and abstracts |
| ArxivClusteringS2S | Identify the main and secondary category of Arxiv papers based on the titles |
| BiorxivClusteringP2P | Identify the main category of Biorxiv papers based on the titles and abstracts |
| BiorxivClusteringS2S | Identify the main category of Biorxiv papers based on the titles |
| MedrxivClusteringP2P | Identify the main category of Medrxiv papers based on the titles and abstracts |
| MedrxivClusteringS2S | Identify the main category of Medrxiv papers based on the titles |
| RedditClustering | Identify the topic or theme of Reddit posts based on the titles |
| RedditClusteringP2P | Identify the topic or theme of Reddit posts based on the titles and posts |
| StackExchangeClustering | Identify the topic or theme of StackExchange posts based on the titles |
| StackExchangeClusteringP2P | Identify the topic or theme of StackExchange posts based on the given paragraphs |
| TwentyNewsgroupsClustering | Identify the topic or theme of the given news articles |
| SprintDuplicateQuestions | Retrieve duplicate questions from Sprint forum |
| TwitterSemEval2015 | Retrieve tweets that are semantically similar to the given tweet |
| TwitterURLCorpus | Retrieve tweets that are semantically similar to the given tweet |
| AskUbuntuDupQuestions | Retrieve duplicate questions from AskUbuntu forum |
| MindSmallReranking | Retrieve relevant news articles based on user browsing history |
| SciDocsRR | Given a title of a scientific paper, retrieve the titles of other relevant papers |
| StackOverflowDupQuestions | Retrieve duplicate questions from StackOverflow forum |
| ArguAna | Given a claim, find documents that refute the claim |
| ClimateFEVER | Given a claim about climate change, retrieve documents that support or refute the claim |
| CQADupstackRetrieval | Given a question, retrieve detailed question descriptions from Stackexchange that are duplicates to the given question |
| DBPedia | Given a query, retrieve relevant entity descriptions from DBPedia |
| FEVER | Given a claim, retrieve documents that support or refute the claim |
| FiQA2018 | Given a financial question, retrieve user replies that best answer the question |
| HotpotQA | Given a multi-hop question, retrieve documents that can help answer the question |
| MSMARCO | Given a web search query, retrieve relevant passages that answer the query |
| NFCorpus | Given a question, retrieve relevant documents that best answer the question |
| NQ | Given a question, retrieve Wikipedia passages that answer the question |
| QuoraRetrieval | Given a question, retrieve questions that are semantically equivalent to the given question |
| SCIDOCS | Given a scientific paper title, retrieve paper abstracts that are cited by the given paper |
| SciFact | Given a scientific claim, retrieve documents that support or refute the claim |
| Touche2020 | Given a question, retrieve detailed and persuasive arguments that answer the question |
| TRECCOVID | Given a query on COVID-19, retrieve documents that answer the query |
| STS* | Retrieve semantically similar text. |
| BUCC/Tatoeba | Retrieve parallel sentences. |
| SummEval | Given a news summary, retrieve other semantically similar summaries |

EXTENDED STUDY

**Noise-robustness is important for dense text encoders.** It is generally assumed that positively paired documents $(x, y^+)$ in dense encoding datasets are correctly aligned, adhering to the rules of some task instruction. However, this assumption is difficult to satisfy in real-world applications (Qu et al., 2021; Wang et al., 2022). In practice, many training data pairs are collected automatically without human inspection, and this inevitably leads to the inclusion of some mismatched pairs. For example, in order to mine positive passages from QA datasets, Karpukhin et al. (2020) declare the highest-ranked passage from BM25 (Robertson & Zaragoza, 2009) that contains the answer as the positive passage. Although it is hard to measure the level of noise without explicitly asking for human annotations, it is very possible that such a process generates false positives by returning passages that do not answer the query at hand even if they have a high lexical match. Furthermore, it is increasingly common to automatically infer a set of hard negatives for each $x$ in the dataset. A standard recipe for mining these is to sample from the top-k documents in a corpus using a retriever, like BM25 in the case of DPR (Karpukhin et al., 2020), or one that is based on a dense encoder, as in the case of ANCE (Xiong et al., 2021) and NV-Retriever (de Souza P. Moreira et al., 2024). While Wang et al. (2023) has clearly demonstrated that the inclusion of hard negatives leads to downstream improvements, these procedures inadvertently introduce false negatives. Qu et al. (2021) examine the top-retrieved passages that were not labeled as positives in the original MSMARCO (Bajaj et al., 2018) dataset, and they find that 70% of them are actually positives. We thus argue that utilizing training objectives that are more robust to noise can lead to downstream improvements in text encoders.

**Intuition for increased robustness to batch size.** We can get intuition about the batch size resilience of BCE losses by analyzing loss gradients. In softmax-based contrastive learning, the denominator, in which the negative pairs appear, can be expressed as a log-sum-exp. This means that each of the $(x_k, y_n)$ pairs' gradient contribution is weighted by each pair's probability to have $y_n$ classified from $x_n$.

$$\frac{\partial}{\partial y_n} \log \sum_{m=1}^{M} \exp\left(s(x_k, y_m)\right) = \frac{\exp\left(s(x_k, y_n)\right)}{\sum_{m=1}^{M} \exp\left(s(x_k, y_m)\right)} \frac{\partial}{\partial y_n} s(x_k, y_n)$$

Combining this with large logit scales $\in (10, 100)$, that are used in order to achieve good downstream performance, we can see that the negative pair with the largest logit $s(x_k, y_{m^*})$ contributes almost all of the gradient, while the rest of the negative pairs with smaller logits $s(x_k, y_m) \leq s(x_k, y_{m^*})$ have significantly smaller gradient contributions. In contrast, BCE loss weights all negative pairs independently from one another.

$$\frac{\partial}{\partial y_n} - \log \sigma\left(-s(x_k, y_n)\right) = \sigma\left(s(x_k, y_n)\right) \frac{\partial}{\partial y_n} s(x_k, y_k)$$

Combining this with large logit scales $\in (10, 100)$ once more, we can see that only the negative pairs that are perceived as positive by the model are going to be used and they will have approximately equal weight ($\approx 1$) among themselves. We argue that this makes the BCE formulation use in-batch negatives more efficiently than softmax-based formulations, as it utilizes simultaneously all erroneous predictions about negative pairs instead of the most erroneous one at each training step.

COMPREHENSIVE RESULTS ON MTEB

Table 7 reports the results with an extra section containing top-5 ranking models in terms of average (Avg) score as of October 1st, 2024. Basing off our study for training with our proposed loss on top of any of these models is an impossible task, and thus comparing head-to-head just by ablating the loss function used for finetuning these general text encoders. The reason is that either the training data has not been made publically available, or the code, or the paper. We choose instead to use one of the models that fulfill all of those open-source requirements. We note that during the time we were developing our work, BGE-EN-ICL had not released a preprint, or their full training data.

Table 8 reports Spearman correlation of gold relevance scores against the predictions from the models for the STS portion of the English MTEB benchmark.

We highlight the performance of our Meta-LLaMA-3 model in the Pair Classification and Semantic Textual Similarity (STS) task categories, where it outperforms models even in the top 5 of the MTEB

Table 7: Results on the 56 tasks of the English MTEB benchmark. Retrieval (Retr.) performance is measure using NDCG@10. Reranking (Rerank.) performance uses Mean Average Precision. Clustering (Clust.) uses the V-measure. Pair classification (Pair) uses Average Precision (AP) based on the models similarity metric. Classification (Class.) uses accuracy. Semantic Textual Similarity (STS) tasks and Summarization (Summ.) use the Spearman correlation based on the model's similarity metric. Finally, the average (Avg) is computed by averaging the scores of all individual tasks together. *versions of models that were finetuned with public MEDI2BGE and E5 datasets.

| Categories → # of datasets → Public? → | Data | Code | Paper | Retr. 15 | Rerank. 4 | Clust. 11 | Pair 3 | Class. 12 | STS 10 | Summ. 1 | Avg 56 |
|---|---|---|---|---|---|---|---|---|---|---|---|
| *Models trained with synthetic, private, or highly overlapping data with MTEB* | | | | | | | | | | | |
| NV-EMBED-V2 [22] | ✓ | ✗ | ✓ | 62.65 | 60.65 | 58.46 | 88.67 | 90.37 | 84.31 | 30.70 | 72.31 |
| BGE-EN-ICL [23] | ✗ | ✓ | ✗ | 62.16 | 59.86 | 58.46 | 88.14 | 88.95 | 84.24 | 30.77 | 71.67 |
| STELLA-EN-1.5B-V5 [link] | ✗ | ✗ | ✗ | 61.01 | 61.21 | 57.69 | 88.07 | 87.63 | 84.51 | 31.49 | 71.19 |
| SFR-EMBEDDING-2-R [27] | ✗ | ✗ | ✗ | 60.18 | 60.14 | 56.17 | 88.07 | 89.05 | 81.26 | 30.71 | 70.31 |
| GTE-QWEN2-7B-INSTRUCT [25] | ✗ | ✗ | ✓ | 60.25 | 61.42 | 56.92 | 85.79 | 86.58 | 83.04 | 31.35 | 70.24 |
| INSTRUCTOR-XL [45] | ✓ | ✓ | ✓ | 49.26 | 57.29 | 44.74 | 86.62 | 73.12 | 83.06 | 32.32 | 61.79 |
| BGE-LARGE-EN-V1.5 [49] | ✓ | ✓ | ✓ | 54.29 | 60.03 | 46.08 | 87.12 | 75.97 | 83.11 | 31.61 | 64.23 |
| GRITLM-7B* [31] | ✗ | ✓ | ✓ | 53.10 | 61.30 | 48.90 | 86.90 | 77.00 | 82.80 | 29.40 | 64.70 |
| E5-MISTRAL-7B-INSTRUCT* [48] | ✗ | ✓ | ✓ | 52.78 | 60.38 | 47.78 | 88.47 | 76.80 | 83.77 | 31.90 | 64.56 |
| ECHO-MISTRAL-7B-INSTRUCT [43] | ✓ | ✓ | ✓ | 55.52 | 58.14 | 46.32 | 87.34 | 77.43 | 82.56 | 30.73 | 64.68 |
| LLM2VEC SHEARED-LLAMA-1.3B | | | | | | | | | | | |
| with softmax-based loss [3] | ✓ | ✓ | ✓ | 51.44 | 55.38 | 43.57 | 86.20 | 72.21 | 83.58 | 30.01 | 61.85 |
| with BCE loss (ours) | ✓ | ✓ | ✓ | 52.28 | 54.16 | 43.59 | 86.61 | 70.56 | 83.10 | 30.79 | 61.58 |
| LLM2VEC META-LLAMA-3-8B-INSTRUCT | | | | | | | | | | | |
| with softmax-based loss [3] | ✓ | ✓ | ✓ | 56.63 | 59.68 | 46.45 | 87.80 | 75.92 | 83.58 | 30.94 | 65.01 |
| with BCE loss (ours) | ✓ | ✓ | ✓ | 57.38 | 59.29 | 47.18 | 88.85 | 76.05 | 85.31 | 31.19 | 65.72 |
| MISTRAL-7B-INSTRUCT | | | | | | | | | | | |
| LLM2VEC with softmax-based loss [3] | ✓ | ✓ | ✓ | 55.99 | 58.42 | 45.54 | 87.99 | 76.63 | 84.09 | 29.96 | 64.80 |
| with BCE loss (ours) | ✓ | ✓ | ✓ | 57.43 | 58.64 | 46.34 | 88.53 | 74.31 | 84.90 | 30.14 | 65.04 |
| QWEN2-7B-INSTRUCT | | | | | | | | | | | |
| with softmax-based loss | ✓ | ✓ | ✓ | 57.67 | 59.62 | 49.37 | 88.67 | 75.85 | 84.95 | 31.30 | 66.13 |
| with BCE loss (ours) | ✓ | ✓ | ✓ | 57.84 | 59.59 | 48.67 | 88.76 | 76.84 | 84.67 | 31.73 | 66.22 |

Table 8: Results on the STS tasks of the English MTEB benchmark. Scores depict Spearman correlation based on the model's similarity metric (usually the cosine similarity).

| Datasets | BIOSSES | SICK-R | STS12 | STS13 | STS14 | STS15 | STS16 | STS17 | STS22 | STSBenchmark | Avg |
|---|---|---|---|---|---|---|---|---|---|---|---|
| NV-EMBED-V2 [22] | 87.42 | 82.15 | 77.89 | 88.30 | 84.30 | 89.04 | 86.77 | 90.67 | 68.12 | 88.41 | 84.31 |
| BGE-EN-ICL [23] | 86.47 | 83.87 | 78.14 | 86.59 | 82.83 | 87.77 | 87.04 | 91.25 | 70.07 | 88.42 | 84.24 |
| STELLA-EN-1.5B-V5 [link] | 83.11 | 82.89 | 80.09 | 89.68 | 85.07 | 89.39 | 87.15 | 91.35 | 68.10 | 88.23 | 84.51 |
| SFR-EMBEDDING-2-R [27] | 87.60 | 77.01 | 75.67 | 82.40 | 79.93 | 85.82 | 84.50 | 88.93 | 67.10 | 83.60 | 81.26 |
| GTE-QWEN2-7B-INSTRUCT | 81.37 | 79.28 | 79.55 | 88.83 | 83.87 | 88.54 | 86.49 | 88.73 | 66.88 | 86.85 | 83.04 |
| INSTRUCTOR-XL [45] | 84.15 | 81.70 | 75.32 | 87.44 | 81.87 | 88.94 | 85.38 | 90.54 | 68.65 | 86.56 | 83.06 |
| BGE-LARGE-EN-V1.5 [49] | 84.65 | 81.68 | 79.05 | 86.37 | 82.78 | 88.03 | 86.49 | 87.50 | 67.05 | 87.52 | 83.11 |
| GRITLM-7B [31] | 86.35 | 83.13 | 77.34 | 85.04 | 82.91 | 88.13 | 86.24 | 90.13 | 68.63 | 85.64 | 83.35 |
| E5-MISTRAL-7B-INSTRUCT [48] | 85.55 | 82.64 | 79.66 | 88.43 | 84.54 | 90.43 | 87.68 | 91.75 | 66.98 | 88.60 | 84.63 |
| ECHO-MISTRAL-7B-INSTRUCT [43] | 86.54 | 83.23 | 76.13 | 83.19 | 80.60 | 87.16 | 85.16 | 90.88 | 67.04 | 85.67 | 82.56 |
| LLM2VEC META-LLAMA-3-8B-INSTRUCT | | | | | | | | | | | |
| with softmax-based loss [3] | 84.92 | 83.94 | 79.27 | 84.83 | 82.94 | 88.09 | 86.54 | 89.58 | 67.67 | 88.05 | 83.58 |
| with BCE loss (ours) | 86.94 | 83.73 | 81.39 | 87.96 | 85.58 | 89.78 | 87.89 | 92.02 | 69.01 | 88.83 | 85.31 |

leaderboard (as of October 1, 2024), even though those models have been trained on synthetic and/or private data (Wang et al., 2023; Muennighoff et al., 2024; Meng et al., 2024), or training data that highly overlaps with the benchmark (Lee et al., 2024; Li et al., 2024). We highlight that our model outperforms all other methods *without being exposed* to data from these categories during training, demonstrating increased robustness compared to models trained with softmax-based contrastive learning. For instance, focusing on STS tasks in Table 8, we find that our model outperforms existing models in 6 out of 10 tasks, leading the average average Spearman correlation of sentence similarities across STS tasks, without compromising its performance in other task categories. We believe that the observed advantages in the Pair Classification and STS tasks is due to the particular form of the binary cross-entropy objective, which aligns more closely with the pairwise similarity objectives in these downstream tasks. This alignment enhances generalization capabilities of models compared to those trained with softmax-based loss.

For completeness of results, we provide in Table 9 the comprehensive list of results of all individual 56 task in the English MTEB for the LLM2Vec models finetuned with our proposed loss.

Table 9: Individual task scores of models trained with BCE loss.

| Task | META-LLAMA-3-8B-INSTRUCT | SHEARED-LLAMA-1.3B |
|---|---|---|
| *Classification* | | |
| AmazonCounterfactualClassification | 80.42 | 79.04 |
| AmazonPolarityClassification | 91.38 | 82.24 |
| AmazonReviewsClassification | 51.53 | 44.91 |
| Banking77Classification | 84.83 | 82.21 |
| EmotionClassification | 53.45 | 46.78 |
| ImdbClassification | 87.50 | 74.87 |
| MassiveIntentClassification | 77.97 | 74.19 |
| MassiveScenarioClassification | 80.14 | 77.74 |
| MTOPDomainClassification | 95.40 | 91.84 |
| MTOPIntentClassification | 78.62 | 66.63 |
| ToxicConversationsClassification | 67.31 | 63.28 |
| TweetSentimentExtractionClassification | 64.03 | 63.01 |
| *Clustering* | | |
| ArxivClusteringP2P | 47.90 | 45.46 |
| ArxivClusteringS2S | 46.32 | 39.07 |
| BiorxivClusteringP2P | 36.93 | 35.41 |
| BiorxivClusteringS2S | 38.28 | 34.04 |
| MedrxivClusteringP2P | 32.52 | 31.92 |
| MedrxivClusteringS2S | 32.83 | 31.41 |
| RedditClustering | 64.31 | 57.44 |
| RedditClusteringP2P | 61.46 | 59.23 |
| StackExchangeClustering | 70.93 | 62.06 |
| StackExchangeClusteringP2P | 33.95 | 32.52 |
| TwentyNewsgroupsClustering | 53.52 | 50.97 |
| *Pair Classification* | | |
| SprintDuplicateQuestions | 96.54 | 95.98 |
| TwitterSemEval2015 | 82.39 | 77.13 |
| TwitterURLCorpus | 87.61 | 86.71 |
| *Reranking* | | |
| AskUbuntuDupQuestions | 65.09 | 59.94 |
| MindSmallReranking | 32.32 | 32.17 |
| SciDocsRR | 86.15 | 77.21 |
| StackOverflowDupQuestions | 53.58 | 47.33 |
| *Retrieval* | | |
| ArguAna | 60.61 | 55.57 |
| ClimateFEVER | 36.96 | 29.05 |
| CQADupstackRetrieval | 43.98 | 39.19 |
| DBPedia | 47.16 | 43.48 |
| FEVER | 92.20 | 89.19 |
| FiQA2018 | 55.85 | 43.56 |
| HotpotQA | 77.12 | 69.33 |
| MSMARCO | 42.83 | 40.91 |
| NFCorpus | 40.09 | 36.01 |
| NQ | 65.44 | 57.04 |
| QuoraRetrieval | 89.98 | 89.42 |
| SCIDOCS | 22.51 | 16.59 |
| SciFact | 77.96 | 69.70 |
| Touche2020 | 24.70 | 24.57 |
| TRECCOVID | 82.47 | 80.57 |
| *STS* | | |
| BIOSSES | 86.94 | 83.85 |
| SICK-R | 83.73 | 82.50 |
| STS12 | 81.39 | 77.04 |
| STS13 | 87.96 | 84.56 |
| STS15 | 89.78 | 88.14 |
| STS14 | 85.58 | 82.12 |
| STS16 | 87.89 | 86.38 |
| STS17 | 92.02 | 91.49 |
| STS22 | 69.01 | 68.45 |
| STSBenchmark | 88.83 | 86.51 |
| *Summarization* | | |
| SummEval | 31.19 | 30.79 |

Table 10: Statistics on the TREC-DL dataset splits with graded relevancies

| Training set | | | | |
|---|---|---|---|---|
| Number of relevances | | 398235 | | |
| Number of queries | | 199 | | |
| Number of documents per query | | | | |
| Min | 25% | 50% | 75% | Max |
| 75 | 168.5 | 227 | 330.5 | 47649 |
| Distribution of relevancies | | | | |
| Relevancy | 0 | 1 | 2 | 3 |
| Probability | 0.731 | 0.143 | 0.119 | 0.007 |

| Validation set | | | | |
|---|---|---|---|---|
| Number of relevances | | 6132 | | |
| Number of queries | | 22 | | |
| Number of documents per query | | | | |
| Min | 25% | 50% | 75% | Max |
| 134 | 191.25 | 243 | 313.25 | 695 |
| Distribution of relevancies | | | | |
| Relevancy | 0 | 1 | 2 | 3 |
| Probability | 0.549 | 0.208 | 0.133 | 0.110 |

| Test set | | | | |
|---|---|---|---|---|
| Number of relevances | | 17818 | | |
| Number of queries | | 82 | | |
| Number of documents per query | | | | |
| Min | 25% | 50% | 75% | Max |
| 114 | 158.25 | 183.50 | 268.50 | 533 |
| Distribution of relevancies | | | | |
| Relevancy | 0 | 1 | 2 | 3 |
| Probability | 0.657 | 0.168 | 0.093 | 0.082 |

TREC-DL DATASET

We use the validation sets from the document retrieval challenge of TREC-DL 2023 competition (Craswell et al., 2024) to define a population of query-document relevancy scores which we will use to define the training and the validation set. In particular, we use 90% of the samples for the training set and 10% for the validation set. Out of those 10%, we only keep those entries whose queries and documents do not overlap with any queries or documents contained in the training set. To create the test set, we use the provided test set from TREC-DL 2023. However, we find that the test set contains some overlap with the set of queries and documents used in the validation sets of the competition. We make sure that the test consists exclusively of held-out data, by removing the entries which correspond to queries and documents found in either of our created training set or validation set. After this procedure, we are left with a query-document dataset with graded relevance scores. The statistics of the dataset are presented at Table 10.

