# OpenReview forum: "Training Universal Text Encoders with Pair Relevance Classification Loss"
_ICLR.cc/2025/Conference — Submitted to ICLR 2025_

### Official Review · Reviewer_tJHr · 2024-10-29

**Soundness:** 3
**Presentation:** 3
**Contribution:** 1
**Rating:** 3
**Confidence:** 5

**Summary:**

The paper adopts a binary classification loss, introduced by SigLIP, to enhance the text embedding learning domain. It follows the same setup as LLM2Vec, with only modifying the original CLIP-style contrastive loss to SigLIP-style binary classification loss. The authors carefully ablate the hyperparameters of this loss and demonstrate that the SigLIP loss brings moderate improvements over CLIP loss on MTEB, STS, and TREC-DL datasets.

**Strengths:**

- Technical sound: The authors conduct experiments on well-established benchmarks, and I verified that the reported baseline numbers match those in the corresponding papers. They also perform extensive ablation studies to explore the impact of SigLIP loss hyperparameters on final performance.
- Paper well written: The paper is generally well-written and easy to understand.

**Weaknesses:**

- Limited contribution: The main concern is the lack of technical novelty. The paper only adapts the SigLIP loss for text embedding learning, strictly following the LLM2Vec setup, with the SigLIP loss as the only modification.
- Minor improvements: The improvements achieved by SigLIP loss are minimal compared to the original CLIP loss.
- Lack of insightful analysis: The paper does not provide in-depth analysis, such as why SigLIP is a preferable loss compared to CLIP loss, why it performs better on MetaLLaMA3 but not ShearedLLaMA3, or why it enhances graded relevance scores in retrieval settings.

**Questions:**

See weaknesses.

---

> ### Author Response · Authors · 2024-11-21
> **Rebuttal 1/**
>
> We thank the reviewer for their constructive comments and positive feedback regarding the technical soundness of our work and clarity of the exposition. Hereby we address their concerns:
>
> ## 1. Clarifying conceptual novelty and real-world implications of the work
>
> - Our work clarifies the link between the logit bias and the label-imbalanced classification problem that is brought by making use of the in-batch negatives; something which is not apparent from the SigLIP paper. This allowed us to derive a formula for setting the correct value for the logit bias per total batch size used, which in turn allows us to train effectively strong generalist text encoding models at a considerable fraction of the batch size typically used for softmax-based contrastive learning approaches. Upon requests of reviewers EFpF and rQBS, please find in Fig.3a experimental evidence for this, also please refer to point 4.1.1 of this rebuttal regarding theoretical intuition for this effectiveness.
>
> - We are demonstrating increased robustness to weakly-supervised dataset noise, in new evidence towards reviewer rQBS. Please find in Fig.3b experimental evidence for it, also please refer to point 4.1.2 of this rebuttal regarding theoretical intuition for the improved robustness.
>
> - The novelty concern at the level discussed by the reviewer can also be applied to the SigLIP paper itself. At the time of SigLIP paper’s review process, binary cross-entropy losses for representation learning have already been explored, as we point out in our extended related work section in the Appendix. To our knowledge, binary cross-entropy loss in representation learning was first explored by [Hjelm et al., 2019]. Their work introduces Deep InfoMax (DIM), which in its simplest form consists of a binary cross-entropy loss and the logits are computed from pairs of views: pairs derived from the same image are labeled as positive, while pairs from different images are labeled as negatives. Following similar reasoning to the reviewer, SigLIP can be thought of as an application of the DIM loss to aligning an image view to a caption “view”. However, this reasoning under-appreciates the successful application of the binary cross-entropy loss to the corresponding domain. Similarly for our work, as pointed out by reviewer EFpF: “This is the first work to demonstrate the pointwise sigmoid loss outperforms the standard contrastive loss for training text embeddings.”.
>
> - Finally, we would like to underline the most distinctive feature of our work: It lays important groundwork for future graded relevance works. Language tasks are more complex and nuanced than just binary labels. For example, although it is common to have only two classes in sentiment classification (positive and negative), in the real-world, the range of sentiments are much more than that. While today there are limited resources available containing graded relevance data, such as the TREC datasets and STS, in the future BCE losses can provide a definite advantage in dense text encoding when one considers the possibility of annotating the data pair relevance with large language generative models. Our work lays the groundwork for such advances by allowing us to encode the uncertainty about a pair of text into the continuum of (0,1). Our work is the first one to successfully exploit the finer grained signal provided by graded relevance in the context of text encoding, as evidence by our experimental results on TREC-DL, and we are looking forward to applying our non-trivial, but simple, training recipe to future work leveraging larger graded relevance datasets.
>
> ## 2. Extending experimental evidence beyond LLM2Vec setup
>
> We have updated the manuscript to include models finetuned from Qwen2-7B-Instruct and Mistral-7B-Instruct. Deviating from the LLM2Vec recipe, we just enabled the bidirectional connection and created a representation by average pooling the token representations which correspond to the embed texts $x$ and $y$, excluding task description or instruction tokens. Both for BCE and softmax-based we are searching for the best batch size and learning rate. We find that both for Mistral-7B and Qwen2-7B training BCE surpasses softmax-based losses on average across the 56-task English MTEB. Please find more details on the updated paper (Table 2) and training details in the Appendix.

---

> > ### Author Response · Authors · 2024-11-21
> > **Rebuttal 2/**
> >
> > ## 3. Addressing the concern about improvements
> >
> > While MTEB results serve as testimony that our method is on par and at times brings minor improvements over softmax-based loss, the focus of our work remains on merits that can be achieved via the BCE loss that cannot be achieved by softmax-based ones. We are here underlining benefits to using BCE loss over softmax which are not minor:
> >
> > ### 3.1 BCE improved performance in smaller batch sizes
> >
> > Given the same computation budget, we can now effectively scale the training of dense text encoder models to larger context lengths, and if needed model sizes. Please refer to Fig. 3a of our paper where we have further solidified our original argument with a theoretical insight in the Appendix. See also rebuttal point 4.1.1.
> >
> > ### 3.2 BCE improved robustness to noise
> >
> > In response to reviewer rQBS, we have further conducted experiments with artificial noise, flipping independently for each query in the training set a positive pair with a hard negative one with some probability. In revised Fig. 3b, we show that BCE loss outperforms softmax-based losses for all noise probabilities in \{0.05, 0.1, 0.2, 0.5\} in terms of average MTEB score. We discuss the intuition behind this in the main text. See also rebuttal point 4.1.2 for our theoretical intuition on why.
> >
> > ### 3.3 BCE has major advantage in using graded relevance data
> >
> > Our experimental results on TREC-DL show that only BCE loss is able to take advantage of the finer grained signal of graded relevance data. Note that in Table 3, we have run our best performing BCE recipe with 5 random seeds, demonstrating small variance and that consistently the BCE loss brings superior results to other training recipes.

---

> > > ### Author Response · Authors · 2024-11-21
> > > **Rebuttal 3/**
> > >
> > > ## 4. Expanding Analysis
> > >
> > > ### 4.1 why SigLIP is a preferable loss compared to CLIP loss
> > >
> > > We have updated our paper to discuss theoretical insights on why we expect BCE losses to perform better in small batch sizes than softmax-based losses, as well as why we expect improved robustness to weakly-supervised dataset noise.
> > >
> > > **4.1.1. Intuition on why BCE performs better in smaller batch sizes than softmax-based**
> > >
> > > We can get intuition about the batch size resilience of BCE losses by analyzing loss gradients. In softmax-based contrastive learning, the denominator, in which the negative pairs appear, can be expressed as a log-sum-exp. This means that each of the $(x_k, y_n)$ pairs' gradient contribution is weighted by each pair’s probability to have $y_n$ classified from $x_n$.
> > > $$
> > > \frac{\partial}{\partial y_n} \log \sum_{m=1}^M \exp \left( s(x_k, y_m) \right) = \frac{\exp\left(s(x_k, y_n)\right)}{ \sum_{m=1}^M \exp \left( s(x_k, y_m) \right)} \frac{\partial}{\partial y_n} s(x_k, y_n)
> > > $$
> > > Combining this with large logit scales $\in (10, 100)$, that are used in order to achieve good downstream performance, we can see that the negative pair with the largest logit $s(x_k, y_{m^*})$ contributes almost all of the gradient, while the rest of the negative pairs with smaller logits $s(x_k, y_m) \leq s(x_k, y_{m^*})$ have significantly smaller gradient contributions. In contrast, BCE loss weights all negative pairs independently from one another.
> > > $$
> > > \frac{\partial}{\partial y_n} - \log\sigma\left(- s(x_k, y_n\right) ) = \sigma\left( s(x_k, y_n) \right) \frac{\partial}{\partial y_n} s(x_k, y_n)
> > > $$
> > > Combining this with large logit scales $\in (10, 100)$ once more, we can see that only the negative pairs that are perceived as positive by the model are going to be used and they will have approximately equal weight ($\approx 1$) among themselves. We argue that this makes the BCE formulation more sample efficient when it comes to using in-batch negatives, as it utilizes simultaneously all erroneous predictions about negative pairs instead of the most erroneous one at each training step.
> > >
> > > **4.1.2. Intuition on why BCE is more robust to noise**
> > >
> > > Suppose the binary label case with hard negatives. At each training step the model encounters a batch of size $B$, the BCE-loss then is the average of losses from $2B^2$ binary classification predictions, while softmax-based loss is the average of losses from $B$ multi-class classification predictions. Assuming that the in-batch negatives are indeed true negatives: If one pair of texts in the batch is mislabeled, then for the softmax-loss this means that the $\frac{1}{B}$ predictions is optimized towards a false target, whereas for the BCE-loss $\frac{1}{B^2}$ predictions is optimized towards a false target. We hypothesize that BCE achieves better robustness to weakly-supervised dataset noise, because the effect of noise is more diluted in the gradients of the BCE objective.
> > >
> > > ### 4.2 why it performs better on MetaLLaMA3 but not ShearedLLaMA3
> > >
> > > As argued in Section 3 of the rebuttal, while MTEB results serve as testimony that our method is on par and at times brings minor improvements over softmax-based loss, the focus of our work remains on merits that can be achieved via the BCE loss that cannot be achieved by softmax-based ones.
> > >
> > > ### 4.3 why it enhances graded relevance scores in retrieval settings
> > >
> > > As argued in Section 3.3 of the paper, training on graded relevance score is straightforward with BCE loss, since we are able to perform essentially regression on the graded values in the continuum (0, 1). Since training explicitly regresses similarity on graded scores, it is expected to generalize better on downstream tasks that evaluate similarity on graded scores, compared to models trained by softmax-based contrastive losses that do not use graded scores.
> > >
> > > ## Proposed revisions
> > >
> > > We believe that the reviewer’s comments have led to a productive rebuttal, allowing us to strengthen the merits of our method and the message of our work, which we believe that it opens interesting and impactful future directions for the text encoding community.
> > >
> > > - Expand theoretical and experimental discussions on noise robustness in the main text.
> > > - New data point for batch size = 2048 for Meta-Llama, as well as theoretical intuition on why BCE is more robust to batch size than softmax-base losses in the Extended Study section of the Appendix.
> > > - Update Table 2 with Qwen2-7B-Instruct and Mistral-7B-Instruct models
> > > - Discuss AIR-Benchmark results on test set to the Appendix
> > >
> > >
> > > **References**
> > >
> > > Hjelm et al., “Learning deep representations by mutual information estimation and maximization”, ICLR 2019

---

> > > > ### Comment · Reviewer_tJHr · 2024-11-25
> > > >
> > > > I appreciate the authors’ detailed response. However, most of my concerns remain unresolved. I have reservations about the new justification for novelty, as well as the limited improvements demonstrated by the newly added models. Additionally, the newly derived theory appears to lack technical rigor. Furthermore, there is still no clear justification for why the proposed method performs well on certain models but not on others. Therefore, I will maintain my original evaluation.

---

### Official Review · Reviewer_rQBS · 2024-11-04

**Soundness:** 3
**Presentation:** 2
**Contribution:** 2
**Rating:** 3
**Confidence:** 3

**Summary:**

This study proposes a new approach for fine-tuning the universal text encoders using binary cross-entropy loss. Using a simple pair classification loss with binary cross-entropy on relevance labels, the proposed method outperforms the standard softmax-based loss, improving average scores in the MTEB benchmark evaluations. Tested on the Meta-Llama-3-8B-Instruct model, this binary cross-entropy approach excels in benchmarks for Pair Classification and STS as well as offering better generalization advantages than softmax-based loss.

**Strengths:**

See the summary.

**Weaknesses:**

In general, the BCE loss has been previous studied in ML application literature as author described in Appendix (extended related work) and the experiments are relatively for small set of models and mteb dataset. In addition, the reported MTEB accuracy number is much lower the top-performing models in the MTEB leaderboard. More comments are included in questions.

**Questions:**

- What is the theoretical reasoning behind the claim that BCE loss outperforms softmax-based loss in fine-tuning embedding models?
- Besides LLM2VEC, can this loss function be used effectively with other embedding models, such as transformer-encoder based models like e5?
- Is this loss advantageous not only for the Llama model but also when applied to other foundational LLMs like Mistral-7b or Qwen-7b?
- The evaluation results are based solely on the MTEB dataset. Since the training data overlaps with MTEB’s training sets, the fine-tuned model might be overfitted to MTEB. Could you provide zero-shot evaluation results, for instance, using the AIR-Benchmark (https://github.com/AIR-Bench/AIR-Bench)?
- What is the theoretical intuition behind the proposed loss’s batch robustness?
- Does incorporating BCE loss also enhance accuracy when non-retrieval datasets (e.g., clustering, classification, STS) are included in the training data?

---

> ### Author Response · Authors · 2024-11-21
> **Rebuttal 1/**
>
> Dear Reviewer,
>
> Thank you for your constructive feedback. We appreciate your insights and would like to address your concerns systematically:
>
> ## Intuition for why BCE is a better loss
>
> We will support this claim by elucidating with theoretical insight two sets of empirical observations: Increased robustness to noise and increased robustness to batch size
>
> ### BCE performs better than softmax loss in smaller batch sizes
>
> **Setting the correct logit bias**
>
> We further solidify this claim in the revised Fig. 3b by providing an additional point for batch sizes 2048 training Meta-LLaMA-3-8B-Instruct. Our new empirical evidence reinforces our original claim, that binary cross-entropy losses improve robustness with respect to batch size because they attain competitive, and at times improved, performance over the softmax-based baseline with smaller batch sizes (256, 512), where performance is almost constant. Crucially in our experiments we are using a logit bias depending on the total batch size used, according to the proposed formula $- \log( 2 B - 1)$ (see Section 3.1, Eq. 3). In Section 3, we discuss the role of logit bias for enabling robust training in label-imbalanced classification scenaria, connecting it to the literature of logit adjustment [Menon et al., 2021]. At the same time, the effect that various values for the logit bias can have in fixed batch-size training are evident in Fig. 2.b.
>
> **Theoretical insight on why is better in smaller batch sizes**
>
> We can get intuition about the batch size resilience of BCE losses by analyzing loss gradients. In softmax-based contrastive learning, the denominator, in which the negative pairs appear, can be expressed as a log-sum-exp. This means that each of the $(x_k, y_n)$ pairs' gradient contribution is weighted by each pair’s probability to have $y_n$ classified from $x_n$.
> $$
> \frac{\partial}{\partial y_n} \log \sum_{m=1}^M \exp \left( s(x_k, y_m) \right) = \frac{\exp\left(s(x_k, y_n)\right)}{ \sum_{m=1}^M \exp \left( s(x_k, y_m) \right)} \frac{\partial}{\partial y_n} s(x_k, y_n)
> $$
> Combining this with large logit scales $\in (10, 100)$, that are used in order to achieve good downstream performance, we can see that the negative pair with the largest logit $s(x_k, y_{m^*})$ contributes almost all of the gradient, while the rest of the negative pairs with smaller logits $s(x_k, y_m) \leq s(x_k, y_{m^*})$ have significantly smaller gradient contributions. In contrast, BCE loss weights all negative pairs independently from one another.
> $$
> \frac{\partial}{\partial y_n} - \log\sigma\left(- s(x_k, y_n\right) ) = \sigma\left( s(x_k, y_n) \right) \frac{\partial}{\partial y_n} s(x_k, y_n)
> $$
> Combining this with large logit scales $\in (10, 100)$ once more, we can see that only the negative pairs that are perceived as positive by the model are going to be used and they will have approximately equal weight ($\approx 1$) among themselves. We argue that this makes the BCE formulation more sample efficient when it comes to using in-batch negatives, as it utilizes simultaneously all erroneous predictions about negative pairs instead of the most erroneous one at each training step.
>
> ### Robustness to weakly-supervised dataset noise
>
> We would like to draw your attention to this point raised by reviewer EFpF, and here we argue about it with experimental evidence and theoretical insight.
>
> **BCE improves robustness to noise compared to softmax loss**
>
> We have performed an ablation study with respect to levels of artificial noise introduced to the open-source E5 training dataset. In particular, we specify a probability of flipping the label between a designated positive pair $(x, y^+)$ and a hard negative pair $(x, y^-)$, and we sample the flip variable independently for every triplet $(x, y^+, y^-)$ in the dataset prior to training. Then, we train Meta-LLaMA-3-8B-Instruct models with the BCE and softmax-based losses and we report the average 56-task MTEB score of the best models found during training. In Fig. 3.b, we observe that BCE outperforms the softmax variant for all noise levels $p_{\text{noise}} \in \{0.05, 0.1, 0.2, 0.5\}$.
>
> **Intuition on why BCE is more robust to noise**
>
> Suppose the binary label case with hard negatives. At each training step the model encounters a batch of size $B$, the BCE-loss then is the average of losses from $2B^2$ binary classification predictions, while softmax-based loss is the average of losses from $B$ multi-class classification predictions. Assuming that the in-batch negatives are indeed true negatives: If one pair of texts in the batch is mislabeled, then for the softmax-loss this means that the $\frac{1}{B}$ predictions is optimized towards a false target, whereas for the BCE-loss $\frac{1}{B^2}$ predictions is optimized towards a false target. We hypothesize that BCE achieves better robustness to weakly-supervised dataset noise, because the effect of noise is more diluted in the gradients of the BCE objective.

---

> > ### Author Response · Authors · 2024-11-21
> > **Rebuttal 2/**
> >
> > ## Generality Across Models
> >
> > We have updated the manuscript to include models finetuned from Qwen2-7B-Instruct and Mistral-7B-Instruct. Deviating from the LLM2Vec recipe, we just enabled the bidirectional connection and created a representation by average pooling the token representations which correspond to the embed texts $x$ and $y$, excluding task description or instruction tokens. Both for BCE and softmax-based we are searching for the best batch size and learning rate. We find that both for Mistral-7B and Qwen2-7B training BCE surpasses softmax-based losses on average across the 56-task English MTEB. Please find more details on the updated paper (Table 2) and training details in the Appendix.
> >
> > ## AIR-Benchmark evaluation
> >
> > We understand the concern about overlap between training data and MTEB tasks. We will soon provide zero-shot evaluations on the QA category of AIR-Benchmark 24.04 for models trained with softmax and BCE objectives. Notice that our models have been finetuned with maximum context length of 512, as they were originally intended only for comparison between the softmax and bce variants of losses.
> >
> > ## Training on more data
> >
> > We repeat that we have purposefully chosen public datasets to ensure reproducibility, and we have made sure to have a minimal overlap between the domains of our training datasets and MTEB so that we can evaluate softmax-based contrastive learning and bce-based contrastive learning, fairly, on downstream zero-shot evaluations. We also underline that our current training set does not only contain retrieval datasets. In particular, the AllNLI portion of our data mix (see Table 1) is also used to train on the STS-type instruction: “Retrieve semantically similar text” (see Table 6). Similar to previous approaches such as LLM2Vec, Echo Embeddings and E5-Mistral, we strive to keep the data overlap with MTEB minimal, to test the true generalization capabilities of the model. Even though we have not explicitly trained on training sets from STS, we would like to stress that our Meta-LLaMA-3-8B-Instruct model’s zero-shot performance is better than models that have trained on STS. NV-Embed-V2 (see Table 3) is a model that was trained on a combination of clustering, classification and STS data from MTEB and it underperforms compared to our Meta-LLaMA-3-8B-Instruct. We consider this as a demonstration of the zero-shot capabilities of our model.
> >
> > Our models performance will improve when further fine-tuned on domain data. One example of this is when we finetune on the TREC-DL graded relevance dataset we have assembled (see Table 4).
> >
> > ## Proposed revisions
> >
> > We are grateful for your thoughtful questions and believe these additions will strengthen our contributions to the field.
> >
> > - Expand theoretical and experimental discussions on noise robustness in the main text.
> > - New data point for batch size = 2048 for Meta-Llama, as well as theoretical intuition on why BCE is more robust to batch size than softmax-base losses in the Extended Study section of the Appendix.
> > - Update Table 2 with Qwen2-7B-Instruct and Mistral-7B-Instruct models
> > - Discuss AIR-Benchmark results on test set to the Appendix
> >
> > **References**
> >
> > Menon et al., Long-tail learning via logit adjustment”, ICLR 2021

---

> ### Comment · Reviewer_rQBS · 2024-11-27
>
> - In Table 2, the results for Sheared-Llama-1.3B indicate that the BCE loss performs worse compared to the softmax-based loss. Other experiments also show only marginal difference in average scores (e.g., less than 0.1 for Qwen2-7B).
> - Regarding Table 2, the literature model does not include the leading models, and this table partially overlaps with the results presented in Table 7.
> - Please include the AIR-Benchmark as a zero-shot evaluation metric to show the effectiveness of the proposed method.
> - Final comment: Thank you for the detailed responses. While the idea of enhanced batch and noise robustness through the loss function is interesting, the experimental results and presentation do not provide sufficient support for this claim. I will maintain my score.

---

### Official Review · Reviewer_EFpF · 2024-11-04

**Soundness:** 2
**Presentation:** 2
**Contribution:** 2
**Rating:** 5
**Confidence:** 5

**Summary:**

This paper proposes pair relevance classification loss for training text embeddings. The pair relevance classification loss generalizes the sigmoid loss of the SigLIP work, and is a binary cross entropy loss between model prediction and ranking labels. When used with binary labels as most text embedding data, the loss reverse back to the same sigmoid loss as proposed in SigLIP. Compared to the standard contrastive learning loss, the loss falls in the category of point-wise ranking, where the model is optimized against absolute ranking labels instead of relative ranking of different documents. Experiments on the MTEB show the proposed loss leads to better embeddings than the standard contrastive loss. Furthermore, pair relevance classification loss can be naturally used when graded relevance labels are available.

**Strengths:**

1. The proposed pair relevance classification loss generalizes the sigmoid loss from SigLIP work.
2. This is the first work to demonstrate the pointwise sigmoid loss outperforms the standard contrastive loss for training text embeddings.
3. The proposed loss can better utilize data where graded labels are available.

**Weaknesses:**

1. The vast majority of existing text training data are of binary labels. For the binary labeled data, the proposed loss reverses back to the sigmoid loss of previous work.
2. For graded labels, this work fails to mention previous work on pointwise ranking which do regression on the absolute relevance labels. For example, see section 2.4 in https://arxiv.org/pdf/2105.11108.
3. The paper does not provide intuitions or analysis why the proposed loss would be better for the binary labeled text data. In the SigLIP work, the postulated benefits are 1) efficient processing enables large batch size in training and 2) robustness against noises prevalent in language-image pretraining data. In this work, the training batch size is relatively small, and it does not demonstrate noise robustness plays any roles in the text domain.

**Questions:**

The proposed loss falls into the category of pointwise ranking loss. Classic work such as Learning to Rank for Information Retrieval (http://didawikinf.di.unipi.it/lib/exe/fetch.php/magistraleinformatica/ir/ir13/1_-_learning_to_rank.pdf) usually consider pointwise approaches inferior to other methods as they lack the global view. As a matter of fact, I personally tried the sigmoid loss to train text embeddings with large language models before I review this paper, but failed to gain any benefit compared to the standard contrastive loss. Can the authors provide any intuitions why the pointwise loss would be better in this case?

---

> ### Author Response · Authors · 2024-11-21
> **Rebuttal 1/**
>
> We appreciate the reviewer’s thoughtful feedback, which has provided valuable insights for improving our work. Below, we address the points raised, highlighting areas of agreement, clarification, and proposed changes.
>
> ## Response to Strengths
>
> - We are glad the reviewer acknowledged the generalization of SigLIP and the utility of graded labels in our proposed approach.
> - The experimental results on MTEB tasks and graded relevance datasets (Section 5) reinforce these observations.
>
> ## Response to Weaknesses
>
> **1. In the case of binary labels, our work clarifies how to set significant hyperparameters making the difference**
>
> While the loss reduces to SigLIP for binary labels, the results in Table 2 demonstrate its superior generalization compared to softmax-based contrastive loss. We are happy to find that the reviewer has already considered experimenting with binary cross-entropy losses for training dense text encoders. This highlights the importance of setting hyperparameters correctly in the training recipe, like the logit bias, which we analyze through the prism of label-imbalance classification. We invite the reviewer to point (3) of our rebuttal for more details.
>
> **Our work is important groundwork for future graded relevance works:**
>
> While we agree with the reviewer that the vast majority of existing text training data are of binary labels, we would like to mention that this happens more out of convenience of data curation. In reality, language tasks are more complex and nuanced than just binary labels. For example, although it is common to have only two classes in sentiment classification (positive and negative), in the real-world, the range of sentiments are much more than that. While today there are limited resources available containing graded relevance data, such as the TREC datasets and STS, in the future BCE losses can provide a definite advantage in dense text encoding when one considers the possibility of annotating the data pair relevance with large language generative models. Our work lays the groundwork for such advances by allowing us to encode the uncertainty about a pair of text into the continuum of (0,1).
>
> **2. Missing comparison to regression-based pointwise ranking:**
>
> We acknowledge this limitation and we have included a discussion of [Zou et al., 2021] in the revised paper. However, we note that our focus is on general-purpose text embeddings, distinct from pointwise ranking models for information retrieval. While the referenced paper mobilizes training objectives on graded relevance labels, it does not prescribe a way to deal with the imbalances of the training relevance data. It is not obvious how to modify their regression-based loss so that it accounts for possible imbalances, nor do they report the marginal distribution of the human annotated relevance data. Our work fills that gap by allowing pointwise training on imbalanced relevance data.

---

> > ### Author Response · Authors · 2024-11-21
> > **Rebuttal 2/**
> >
> > **3. Intuitions for Performance on Binary Data:**
> >
> > **Setting the correct logit bias means better performance with smaller batch sizes**:
> >
> > SigLIP allows for a block-wise processing of the logit matrix, which tradeoffs memory consumption with compute time per training step, enabling the use of larger batch sizes. However, using larger batch sizes does not necessarily lead to downstream performance improvements. This is evident from SigLIP’s Fig. 2b where downstream zero-shot top-1 accuracy follows an inverse U-curve, and our Fig. 3 where we have provided the additional point for batch size 2048 training Meta-LLaMA-3-8B-Instruct. Our range of experimented batch sizes are comparable to other LLM-based text encoders in the literature - 512 for LLM2Vec, 2048 for E5-Mistral, and 2048 for GritLM. Our new empirical evidence reinforces our original claim, that binary cross-entropy losses improve robustness with respect to batch size because they attain competitive, and at times improved, performance over the softmax-based baseline with smaller batch sizes (256, 512), where performance is almost constant.
> >
> > **Setting the correct logit bias means better performance with smaller batch sizes**:
> >
> > SigLIP allows for a block-wise processing of the logit matrix, which tradeoffs memory consumption with compute time per training step, enabling the use of larger batch sizes. However, using larger batch sizes does not necessarily lead to downstream performance improvements. This is evident from SigLIP’s Fig. 2b where downstream zero-shot top-1 accuracy follows an inverse U-curve, and our Fig. 3 where we have provided the additional point for batch size 2048 training Meta-LLaMA-3-8B-Instruct. Our range of experimented batch sizes are comparable to other LLM-based text encoders in the literature - 512 for LLM2Vec, 2048 for E5-Mistral, and 2048 for GritLM. Our new empirical evidence reinforces our original claim, that binary cross-entropy losses improve robustness with respect to batch size because they attain competitive, and at times improved, performance over the softmax-based baseline with smaller batch sizes (256, 512), where performance is almost constant.
> >
> > Notice that in our experiments we are using a logit bias depending on the total batch size used according to the proposed formula $- \log( 2 B - 1)$ (see Section 3.1, Eq. 3). In contrast, SigLIP experiments make the following ablation study with a fixed logit bias of $-10$ and as a result their performance in lower batch sizes is suboptimal (as evidenced by the logit bias ablation in our Fig. 2b). It is noteworthy to observe that their performance peaks exactly at the batch size which produces the logit bias value $-10$ according to our formula. Looking at SigLIP’s Fig. 2b, downstream zero-shot accuracy is maximized at $B=32768$, for which the proposed logit bias would be $- \log B - 1 = - \log 32767 \approx - 10.397$. Such an analysis on the role of logit bias is absent from the SigLIP paper, and it is crucial for achieving the maximum possible performance attained with binary cross-entropy losses; which might be the reason why the reviewer was not able to achieve competitive performance when they tried the BCE loss for training dense text encoders.
> >
> > **Intuition on why BCE is better in smaller batch sizes:**
> >
> > We can get intuition about the batch size resilience of BCE losses by analyzing loss gradients. In softmax-based contrastive learning, the denominator, in which the negative pairs appear, can be expressed as a log-sum-exp. This means that each of the $(x_k, y_n)$ pairs' gradient contribution is weighted by each pair’s probability to have $y_n$ classified from $x_n$.
> > $$
> > \frac{\partial}{\partial y_n} \log \sum_{m=1}^M \exp \left( s(x_k, y_m) \right) = \frac{\exp\left(s(x_k, y_n)\right)}{ \sum_{m=1}^M \exp \left( s(x_k, y_m) \right)} \frac{\partial}{\partial y_n} s(x_k, y_n)
> > $$
> > Combining this with large logit scales $\in (10, 100)$, that are used in order to achieve good downstream performance, we can see that the negative pair with the largest logit $s(x_k, y_{m^*})$ contributes almost all of the gradient, while the rest of the negative pairs with smaller logits $s(x_k, y_m) \leq s(x_k, y_{m^*})$ have significantly smaller gradient contributions. In contrast, BCE loss weights all negative pairs independently from one another.
> > $$
> > \frac{\partial}{\partial y_n} - \log\sigma\left(- s(x_k, y_n\right) ) = \sigma\left( s(x_k, y_n) \right) \frac{\partial}{\partial y_n} s(x_k, y_n)
> > $$
> > Combining this with large logit scales $\in (10, 100)$ once more, we can see that only the negative pairs that are perceived as positive by the model are going to be used and they will have approximately equal weight ($\approx 1$) among themselves. We argue that this makes the BCE formulation use in-batch negatives more efficiently than softmax-based formulations, as it utilizes simultaneously all erroneous predictions about negative pairs instead of the most erroneous one at each training step.

---

> > > ### Author Response · Authors · 2024-11-21
> > > **Rebuttal 3/**
> > >
> > > **Noise-robustness is important for dense text encoders**:
> > >
> > > We disagree with the reviewer on the point that robustness to noise is not relevant for training dense text encoders. Generally, it is assumed that the positively paired documents $(x, y^+)$ in dense encoding datasets are correctly aligned, adhering to the rules of some task instruction. However, this assumption is difficult to satisfy in real-world applications [Qu et al., 2021; Wang et al., 2022]. In practice, many training data pairs are collected automatically without human inspection, and this inevitably leads to the inclusion of some mismatched pairs. For example, in order to mine positive passages from QA datasets, Karphukhin et al. declare the highest-ranked passage from BM25 that contains the answer as the positive passage. Although it is hard to measure the level of noise without explicitly asking for human annotations, it is very possible that such a process generates false positives by returning passages that do not answer the query at hand even if they have a high lexical match. In an ongoing effort to quantify the levels of noise for the training set we have used (open portion of E5), we have asked human annotators to evaluate whether a pair of texts is positive or negative based on whether the dataset task’s instruction is strictly followed. In this process, an annotator is exposed to a pair of texts which 50% of the time are positive according to the dataset label, and 50% are negative. We summarize our ongoing empirical findings about the positively labeled pairs of data below:
> > >
> > > | dataset | #\{human finds it negative\}/#\{dataset positive pairs seen\} |
> > > | :--------- | :-----------------------------------------------------------------: |
> > > | NQ    |        1/7  |
> > > | allnli | 0/12 |
> > > | msmarco\_document | 1/6 |
> > > | msmarco\_passage | 1/7 |
> > > | squad | 1/9 |
> > > | trivia\_qa | 7/21 |
> > > | fever | 0/10 |
> > > | hotpot\_qa | 2/9 |
> > > | quora\_duplicates | 0/7 |
> > >
> > > Furthermore, it is increasingly common to automatically infer a set of hard negatives for each $x$ in the dataset. A standard recipe for mining these is to sample from the top-k documents in a corpus using a retriever, like BM25 in the case of DPR [Karpukhin et al., 2020], or one that is based on a dense encoder, as in the case of ANCE [Xiong et al., 2021] and NV-Retriever [Moreira et al., 2024]. While Wang et al., 2023 has clearly demonstrated that the inclusion of hard negatives leads to downstream improvements, these procedures inadvertently introduce false negatives. Qu et al., 2021 examine the top-retrieved passages that were not labeled as positives in the original MSMARCO dataset, and they find that 70% of them are actually positives. We thus argue that utilizing training objectives that are more robust to noise can lead to downstream improvements in text encoders.
> > >
> > > **BCE improves robustness to noise compared to softmax loss:**
> > >
> > > We have performed an ablation study with respect to levels of artificial noise introduced to the open-source E5 training dataset. In particular, we specify a probability of flipping the label between a designated positive pair $(x, y^+)$ and a hard negative pair $(x, y^-)$, and we sample the flip variable independently for every triplet $(x, y^+, y^-)$ in the dataset prior to training. Then, we train Meta-LLaMA-3-8B-Instruct models with the BCE and softmax-based losses and we report the average 56-task MTEB score of the best models found during training. In Fig. 3.b, we observe that BCE outperforms the softmax variant for all noise levels $p_{\text{noise}} \in \{0.05, 0.1, 0.2, 0.5\}$.
> > >
> > > **Intuition on why BCE is more robust to noise:**
> > >
> > > Suppose the binary label case with hard negatives. At each training step the model encounters a batch of size $B$, the BCE-loss then is the average of losses from $2B^2$ binary classification predictions, while softmax-based loss is the average of losses from $B$ multi-class classification predictions. Assuming that the in-batch negatives are indeed true negatives: If one pair of texts in the batch is mislabeled, then for the softmax-loss this means that the $\frac{1}{B}$ predictions is optimized towards a false target, whereas for the BCE-loss $\frac{1}{B^2}$ predictions is optimized towards a false target. We hypothesize that BCE achieves better robustness to weakly-supervised dataset noise, because the effect of noise is more diluted in the gradients of the BCE objective.

---

> > > > ### Author Response · Authors · 2024-11-21
> > > > **Rebuttal 4/**
> > > >
> > > > ## Response to Questions
> > > >
> > > > The textbook on “Learning to Rank for Information Retrieval” does not discuss pointwise approaches from the robust learning perspective. That is, in the notation of the book, the pointwise learning algorithms exposed are the same, regardless of the distribution of labels $p_\text{data}(y)$ in the training data, where $y$ is a label. However, we would agree that the decision on the relevance of a query to a document should not depend on $p_\text{data}(y)$. This is not the case when we simply model $p_\text{data}(y|x)$ using regression or classification objectives. Special care needs to be taken so that the model learns to be invariant to $p_\text{data}(y)$, in other words predicting according to the invariant $\arg\max_y p(x|y)$. In our work, we attempt to solve this problem by intervening to the training dynamics via setting the logit bias to a particular value. The discrepancy in performance between listwise and pointwise methods may lie in this very consideration. As we show in our work, when the logit bias is handled properly our BCE loss outperforms or competes with the softmax-based one. As long as one is optimizing correctly (using the proper logit bias) using sufficiently many negative pairs (via in-batch negatives), the “global-view” is exposed by optimizing with BCE as well.
> > > >
> > > > ## Proposed Revisions
> > > >
> > > > - Discuss regression-based pointwise ranking method in the extended related work section.
> > > > - Expand theoretical and experimental discussions on noise robustness in the main text.
> > > > - New data point for batch size = 2048 for Meta-Llama, as well as theoretical intuition on why BCE is more robust to batch size than softmax-base losses in the Extended Study section of the Appendix.
> > > >
> > > > Thank you for your constructive comments, which will help us refine our contribution.
> > > >
> > > > ## References
> > > >
> > > > [Zou et al., 2021] Zou et al., “Pre-trained Language Model based Ranking in Baidu Search”, KDD 2021
> > > > [SigLIP] Zhai et al., “Sigmoid Loss for Language Image Pre-Training”, ICCV 2023
> > > > [Qu et al., 2021], Qu et al., “RocketQA: An Optimized Training Approach to Dense Passage Retrieval for Open-Domain Question Answering”, NAACL 2021
> > > > [Wang et al., 2022], Wang et al., “Text Embeddings by Weakly-Supervised Contrastive Pre-training”, arXiv Dec. 2022
> > > > [Wang et al., 2023], Wag et al., “Improving text embeddings with large language models”, arXiv 2023
> > > > [Karpukhin et al., 2020] Karpukhin et al., “Dense Passage Retrieval for Open-Domain Question Answering”, EMNLP 2020
> > > > [Xiong et al., 2021] Xiong et al., “Approximate Nearest Neighbor Negative Contrastive Learning for Dense Text Retrieval“, ICLR 2021
> > > > [Moreira et al., 2024] Moreira et al., “NV-Retriever: Improving text embedding models with effective hard-negative mining“, arXiv Jul. 2024

---

> > > > > ### Comment · Reviewer_EFpF · 2024-11-22
> > > > > **Thanks for the detailed response.**
> > > > >
> > > > > The rebuttal answered my questions about the intuitions on why the BCE loss is better at handling small batch sizes and noises in the data compared to the standard approach. The explanation sounds plausible, but nevertheless has to be verified through experiments. As I mentioned in my original review, I have done embedding experiments with sigmoid loss under similar settings before, and I rerun some of those experiments with calibrated logit bias as suggested in the paper but failed to see any gains compared to the standard baseline.
> > > > >
> > > > > Overall, the author has addressed some of my concerns and I raise my score from 3 to 5.

---

### Official Review · Reviewer_UP2A · 2024-11-07

**Soundness:** 2
**Presentation:** 3
**Contribution:** 2
**Rating:** 5
**Confidence:** 4

**Summary:**

This paper introduces a method to leverage soft relevance scores for text representation learning, addressing limitations in the standard CLIP-style contrastive learning framework. The authors argue that CLIP-style training cannot effectively incorporate relevance scores due to structural constraints in the framework. To address this, they propose using an instance-to-instance binary cross-entropy (BCE) loss with relevance scoring and a logit bias, which they claim can better utilize soft relevance scores without being affected by batch size or long-tail label distributions—issues that can impact traditional CLIP-style training.

**Strengths:**

1. The paper addresses a significant challenge in contrastive learning: that instance pairs often exhibit relationships between purely positive or negative. By accommodating these "in-between" relevance scores, the method could offer a more nuanced approach to contrastive learning.
2. The core idea of the paper is presented clearly.

**Weaknesses:**

1. The paper asserts that measuring relevance with softmax-based contrastive learning is challenging; however, existing soft-contrastive methods, such as IW softmax used in the baseline, enable weighting instance pairs by relevance. A discussion comparing this approach to soft-contrastive learning would be beneficial.
2. The title emphasizes learning a "universal" encoder, but the paper does not provide specific methods geared toward universal representation learning, which could mislead readers.
3. The main contribution—incorporating relevance scores with BCE loss for contrastive learning—yields only modest performance gains over the softmax loss, which may limit the overall impact of the contribution.
4. While the design is conceptually sound, the function of the logit bias β feels somewhat ad hoc. Its purpose resembles methods aimed at addressing long-tail distributions, and related methods should be discussed to situate this approach within the broader context.

Typo:
Line 303: We propose using Suppose we have...

**Questions:**

Please see the weaknesses.

---

> ### Author Response · Authors · 2024-11-21
> **Rebuttal 1/**
>
> Thank you for your review and feedback. We are grateful for your recognition of the significance of our work, accommodating graded relevance scores in contrastive learning, and the clarity of our presentation. Below, we address your comments in detail:
>
> ## 1. About IW softmax
>
> **Softmax-based graded relevance handling (e.g., IW softmax) is an ablation attempt from us.** We would like to stress that IW softmax is not a studied method from the literature. Instead, we introduce it as a method to trivially adapt softmax-based losses for graded relevance scores. Our results demonstrate that this adaptation is difficult, thus justifying the need of our proposed loss. In L507-508: “By this baseline, we want to demonstrate an example that attempts to incorporate graded relevance scores into the standard contrastive loss.”. We have changed the main text to further clarify this.
>
> ## 2. About the term Universal Text Encoders
>
>  The term “Universal” characterizes “Text Encoders” in the title, referring to the fact that we are training text encoders to perform well on a variety of downstream tasks and categories. Our text encoders are benchmarked against a broad spectrum of text related downstream tasks using MTEB benchmark: classification, clustering, pair classification, retrieval, reranking, summarization and semantic textual similarity (STS) tasks. Our text encoders are universal in the sense that we would like them to give a good vector representation for all embedding tasks. This is in contrast to focusing only on just one task, like dense encoders for retrieval for instance.
>
> ## 3. Addressing the concern about improvements
>
> While MTEB results serve as testimony that our method is on par and at times brings minor improvements over softmax-based loss, the focus of our work remains on merits that can be achieved via the BCE loss, that cannot be achieved by softmax-based ones. We are here underlining benefits to using BCE loss over softmax which are not minor:
>
> ### 3.1 Improved performance with smaller batch sizes:
>
> Given the same computation budget, we can now effectively scale the training of dense text encoder models to larger context lengths, and if needed model sizes. Please refer to Fig. 3a of our paper. We have further solidified our original argument with a theoretical insight in the Section “Extended Study” of the Appendix.
>
> ### 3.2 Improved robustness to noise
>
> In response to reviewer rQBS, we have further conducted experiments with artificial noise, flipping with some probability, independently for each query in the training set, the positive pair with the hard negative one. In Fig. 3b of the updated manuscript, we show that BCE loss outperforms softmax-based losses for all noise probabilities in \{0.05, 0.1, 0.2, 0.5\} in terms of average MTEB score. We further discuss the intuition behind this in the main text.
>
> ### 3.3 Major advantage in using graded relevance data
>
> Our experimental results on TREC-DL show that only BCE loss is able to take advantage of the finer grained signal of graded relevance data. Note that in Table 3, we have run our best performing BCE recipe with 5 random seeds, demonstrating small variance and that the BCE loss consistently brings superior results to other training recipes.
>
> ## 4. On the role of logit bias
>
> In Section 3.2 on the role of logit bias, we connect with Menon et al., 2021 and the literature of logit adjustment for long-tail learning. In particular, Menon et al. modify the unnormalized logits of a K-class classifier by adding a fixed estimate for the log-likelihood of each one of the classes to the corresponding class logit predicted by the network. By training this logit-adjusted classifier with the regular cross-entropy loss and removing the logit bias at the end of training, they show that the remaining trained classifier is robust to shifts in the label distribution. We have modified the main text to clarify this connection.
>
> ## 5. Typo
>
> We thank the reviewer for spotting the oversight and we correct it in the revision.
>
> ## Proposed revisions
>
> We appreciate your suggestions and we are confident that they will strengthen the final version of our work.
>
>  - Clarify that IW softmax is a baseline that we propose, not the literature, for attempting an adaptation of softmax-based losses to graded relevance data.
>  - Discuss logit adjustment literature for multi-class imbalanced classification more extensively.
>  - Fix the typo suggested by the reviewer.
>
> ## References
>
> Menon et al., “Long-tail learning via logit adjustment”, ICLR 2021

---

### Meta-Review · Area_Chair_kULQ · 2024-12-19

**Metareview:**

This paper introduces a new loss function for training text embeddings, aiming to improve upon the standard CLIP approach. The core idea is to use a binary cross-entropy (BCE) loss based on pairwise relevance scores between text samples, sometimes referred to as "pair relevance classification loss," which generalizes the sigmoid loss used in SigLIP.  This approach treats embedding training as a pointwise ranking task, optimizing against absolute relevance labels rather than relative rankings used in contrastive learning.   Experiments on the MTEB and other STS tasks show the proposed loss leads to better embeddings than the standard contrastive loss.

Reviewers agree the paper is presented clearly and technically sound. And it is not explored previously applying the pointwise sigmoid loss outperforms the standard contrastive loss for training text embeddings. The major concern, however, is still the novelty. The paper mainly adopts the SigLIP loss into the text encoder training, strictly following the LLM2Vec setup. This limits the overall impact of the contribution. In addition, reviewers concerned about the improvement of using the BCE loss compared to original contrastive loss, and lack of detailed analysis.

**Additional Comments On Reviewer Discussion:**

The reviewers reviewed the authors response. Some of the concerns were addressed by the authors, but the major concern of the novelty still remains after the discussion period.

---

### Decision · Program_Chairs · 2025-01-22

Reject